# Explainable time-series forecasting with sampling-free SHAP for Transformers

Matthias Hertel ⊕ ✉, Sebastian Pütz ⊕, Ralf Mikut ⊕, Veit Hagenmeyer ⊕ & Benjamin Schäfer ⊕

Time-series forecasting is essential for planning and decision-making across domains, and model explainability is critical for fostering user trust and satisfying transparency requirements. We introduce SHAPformer, an accurate, fast and explainable time-series forecasting model based on the Transformer architecture and Shapley Additive Explanations (SHAP). SHAPformer leverages attention manipulation to make predictions using feature subsets, thereby eliminating the need for sampling from background data required by established SHAP algorithms. As a result, it produces exact explanations in less than one second, achieving speedups of 50–1000 × compared to Permutation-SHAP. On synthetic data with known ground-truth explanations, SHAPformer generates explanations that are true to the data. When applied to electrical load data and electricity price data, it achieves competitive predictive performance while providing meaningful local and global insights, including the identification of the past target as the key predictor and the detection of distinct load forecasting behavior during the Christmas period.

Time-series forecasting plays a critical role in many domains, including logistics, retail, finance, healthcare, business services, traffic management, and energy[1]. In the energy sector, forecasts are becoming increasingly important due to the ongoing transition to renewable energy sources[2], which is essential to mitigate the severe impacts of anthropogenic climate change[3]. The inherent variability of renewable generation makes accurate forecasts of electrical load and generation crucial to maintaining the real-time balance between supply and demand[4]. In addition, forecasts enable early detection of critical grid conditions and help prevent equipment overloads through demand-side response and redispatch measures[5].

Across domains, modern forecasting models increasingly rely on complex deep learning models[6,7], including Transformer models tailored to specific use cases[8,9] and Transformer-based Time-Series Foundation Models[10]. These models involve a large number of parameters, rendering their internal processes opaque to human users. This black-box characteristic of machine learning and deep learning models has driven the development of Explainable Artificial Intelligence (XAI) methods, which aim to improve the understanding of how

these models operate[11]. Such model interpretability is valuable for several reasons. First, it can improve user trust and facilitate the adoption of deep learning in real-world applications[12]. Second, explanations help users assess the reliability of predictions and inform decision-making[13]. Third, they assist developers in debugging and refining models – for example, by revealing spurious patterns or "Clever Hans" effects[14]. Finally, XAI supports compliance with regulations demanding transparency in AI systems. The European Union's AI Act, for instance, mandates transparency for applications involving humans or critical infrastructure, such as energy systems[15]. Although not always required to meet regulatory standards, XAI can support human oversight and improve transparency[13], and may justify future regulatory standards for transparency in AI systems.

Shapley Additive Explanations (SHAP)[16] is an XAI framework based on Shapley values from cooperative game theory[17]. SHAP gained popularity because it enables the computation of both feature importance and feature contributions, while satisfying the efficiency property, i.e., the feature contributions sum to the model prediction. In contrast, other post-hoc XAI methods, including LIME[18], Grad-CAM[19]

Institute for Automation and Applied Informatics (IAI), Karlsruhe Institute of Technology (KIT), Eggenstein-Leopoldshafen, Germany.
✉e-mail: matthias.hertel@kit.edu

and Layer-wise Relevance Propagation (LRP)[20,21], either highlight important inputs without indicating their effect on the prediction or do not satisfy the efficiency property. SHAP is the most widely used XAI framework in the energy sector[22] and has been applied to explain forecasts by a variety of models[23], including multilayer perceptrons[24], tree-based methods[25–28], long short-term memory (LSTM) networks[29–31] and Informer models[32].

The core idea of SHAP is to compute the average marginal contribution of a feature across all feature subsets (also called "coalitions"). Since most models do not natively support inputs with absent features, several algorithms have been proposed to estimate SHAP values, including three approaches tailored to time-series models[33–35]:

- KernelSHAP[16] estimates SHAP values by fitting a weighted local linear model to outputs obtained from perturbed samples, where absent features are sampled from background data.
- PermutationSHAP[16] estimates SHAP values by repeatedly permuting features and computing marginal feature contributions, with absent features sampled from background data.
- TreeSHAP[36] is designed for tree-based models and exploits the tree structure to compute SHAP values efficiently.
- DeepSHAP[16] estimates SHAP values for neural networks by applying DeepLIFT[37] contribution rules repeatedly for a set of background samples.
- GradientSHAP is a method for differentiable models based on Expected Gradients[38]. It estimates SHAP values by averaging gradients evaluated at random points between the instance and randomly sampled background samples.
- TimeSHAP[33] is designed for time-series data and estimates SHAP values for selected time steps and features by using KernelSHAP, where absent values are replaced with a baseline.
- WindowSHAP[34] and ShapTime[35] are SHAP algorithms for time series that estimate SHAP values over temporal windows instead of individual time steps, thereby reducing the number of model evaluations.

Most SHAP algorithms follow one of two strategies to evaluate models on feature subsets[39]: (1) sampling absent features from a marginal or conditional distribution, or (2) replacing absent features with a predefined baseline value. Both approaches have limitations. Sampling is computationally intensive and may generate unrealistic inputs that lie outside the data distribution, resulting in off-manifold evaluations[40]. Baseline substitution requires careful selection of the baseline and only yields explanations relative to this baseline.

In addition to model-agnostic explanation methods like SHAP, there are XAI methods specifically designed for Transformer models. Such methods often rely on attention weights as proxies for feature importance[41,42]. However, this practice remains controversial, as the interpretability of attention mechanisms is debated[43–45]. The Temporal Fusion Transformer (TFT)[46] is a specialized Transformer model that computes feature importance values with a dedicated feature selection layer. It has been applied to analyze relevant features in wind power forecasting[47] and frequency forecasting[48], but unlike SHAP, it does not provide information about the feature effects on the prediction. Attention manipulation[49] restricts the Transformer's access to individual features and evaluates the resulting effect on the model prediction, but it does not satisfy the efficiency axiom.

Despite the growing adoption of Transformer architectures for time-series forecasting, explaining their predictions remains challenging. While SHAP is widely used for model interpretability in the energy domain, existing SHAP algorithms are computationally expensive and rely on sampling or baseline substitutions that may produce unrealistic inputs. As a result, there is currently no efficient method for computing SHAP-based feature contributions for time-series Transformers[23]. This limitation motivates the development of approaches that can provide faithful SHAP explanations for such models without relying on sampling or predefined baselines.

To address the need for accurate and explainable methods, we present SHAPformer, a Transformer-based time-series forecasting model that allows to compute exact SHAP values efficiently. An overview of the approach is given in Fig. 1. SHAPformer groups the past target into daily windows and uses one window per exogenous variable, reducing the number of coalitions compared to treating each feature individually. It leverages attention manipulation[49] for training and inference on masked inputs, so that the model can be evaluated on subsets of feature groups without sampling or replacement by baseline values. Our contributions are as follows:

1. We introduce SHAPformer, a time-series forecasting model that combines the strong predictive performance of Transformers with the explainability of SHAP.
2. We validate SHAPformer's explanations on a synthetic dataset with known ground-truth explanations, demonstrating its ability to accurately capture the underlying patterns.
3. We apply SHAPformer to electrical load data from a transmission system operator (TSO) and day-ahead electricity price data from a market bidding zone, showing that SHAPformer generates explanations in less than one second and provides meaningful local and global insights.

## Results

We use three datasets throughout the experiments – a synthetic dataset with known ground-truth explanations for validating

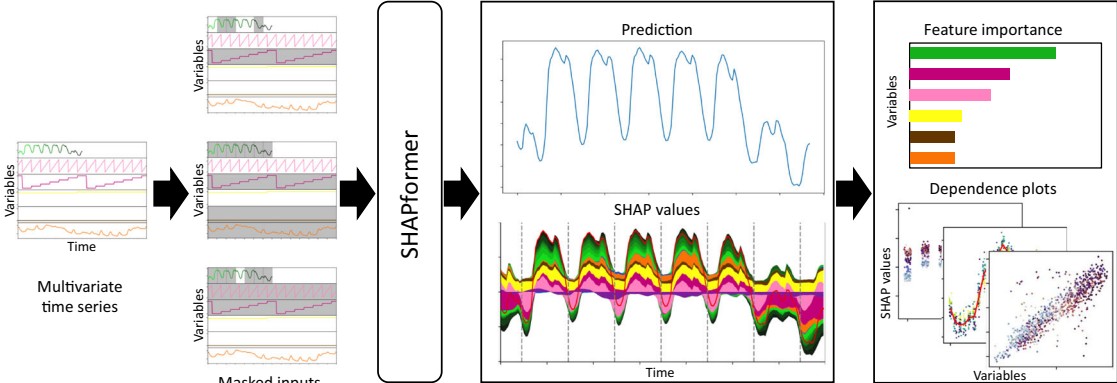

**Fig. 1 | Overview of the proposed method.** The input consists of the past target and covariate time series. The input is divided into feature groups as indicated by colors. SHAPformer makes predictions based on masked inputs. SHAP values are derived from the marginal contributions, defined as the difference of the prediction with and without a feature group. Multiple local explanations are combined into global feature importance values and feature dependence plots.

SHAPformer's explanations, and two empirical datasets to demonstrate its applicability in real-world use cases:

- Synthetic dataset: This dataset exhibits daily, weekly, and annual seasonality with dependencies on exogenous covariates (holidays and a multiplier), for which ground-truth explanations are available. The details of the data generation process are provided in Synthetic dataset. The dataset comprises 100,000 training samples and 10,000 samples each for validation and testing.
- Load dataset: This dataset contains hourly electrical load measurements from the German TSO TransnetBW for the years 2015–2019[50], together with weather data (outside temperature and precipitation) from the Copernicus ERA5 reanalysis model[51]. The first half of 2019 is used for validation, and the second half for testing. The load data is standardized to a zero mean and unit variance.
- Price dataset: This dataset contains day-ahead electricity prices from the DE/LU bidding zone, collected from the ENTSO-E Transparency Platform[52] at hourly resolution for October 2020 until December 2025, together with weather data (wind speed and solar irradiance) from the Copernicus ERA5 reanalysis model[51]. The year 2024 is used for validation, and the year 2025 for testing. The price data is standardized to a zero mean and unit variance.

The temporal split of the datasets is chosen to avoid data leakage and to mimic real-world settings, where models are trained on historical data and evaluated on unseen future observations. SHAPformer is compared against a standard Transformer model explained by three SHAP algorithms – KernelSHAP, PermutationSHAP and WindowSHAP – as well as state-of-the-art forecasting models (detailed in Methods). All models use one week of context and forecast the subsequent 168 h values. Forecast accuracy, training time, and inference time for all models are summarized in Table 1. For the empirical datasets, results are averaged over five runs with different random initializations.

Additional metrics and standard deviations across runs are reported in the Supplementary Results.

## Fast calculation of exact SHAP values while maintaining forecast quality

We compare SHAPformer's predictive performance and its explanations with several baseline forecasting and XAI methods. For the predictive performance evaluation, we use a standard Transformer[53] (as SHAPformer is based on the Transformer architecture), a persistence baseline, linear regression and Extreme Gradient Boosting (XGBoost)[54]. For the explanation comparison, we use TFT and the Transformer explained by three SHAP algorithms: (1) KernelSHAP, which estimates SHAP values by fitting a linear model on predictions based on perturbed inputs; (2) PermutationSHAP, which perturbs inputs using background data under the assumption of feature independence and estimates SHAP values based on the resulting model outputs; and (3) a variant of WindowSHAP, which splits the past target into seven windows (one for each of the preceding seven days) and uses one additional window per exogenous variable, computing SHAP values per window, instead of per feature, by jointly sampling all features within the same window.

The runtime of different XAI methods for explaining a single forecast is reported in Table 2 (evaluated on a machine with an AMD EPYC 7402P processor, 128 GB RAM and an NVIDIA 3090 RTX GPU with 24 GB VRAM). SHAPformer generates explanations more than 50 × faster than PermutationSHAP on the synthetic dataset and 800–1000 × faster on the electricity price and load datasets, achieving runtimes of less than one second per explanation. Remarkably, SHAPformer achieves this speedup despite computing exact SHAP values based on all coalitions of feature groups, whereas PermutationSHAP approximates SHAP values using only ten feature permutations. SHAPformer's speedup comes partly from feature grouping, as WindowSHAP achieves a 134–143 × speedup using the same strategy. In

### Table 1 | Comparison of forecast models

| Model | Forecast error (RMSE) | | | Training time | | | Runtime/forecast | | |
|---|---|---|---|---|---|---|---|---|---|
| | Synthetic – | Load [MW] | Price [€/MWh] | Synthetic [h] | Load [h] | Price [h] | Synthetic [ms] | Load [ms] | Price [ms] |
| Persistence baseline | 0.152 | 652.3 | 49.51 | – | – | – | 0.6 | 1.2 | 1.2 |
| Linear Regression | 0.149 | 553.7 | 40.31 | **0.00** | **0.00** | **0.00** | **0.0** | 0.3 | **0.0** |
| XGBoost | 0.119 | 387.0 | 47.98 | 0.01 | **0.00** | **0.00** | **0.0** | **0.1** | **0.0** |
| TFT | **0.059** | 390.8 | 34.52 | 5.20 | 1.84 | 0.16 | 8.4 | 13.3 | 5.2 |
| Transformer | **0.059** | **263.1** | 35.79 | 0.90 | 0.26 | 0.18 | 11.9 | 4.7 | 5.9 |
| SHAPformer | 0.060 | 265.9 | **33.00** | 10.55 | 3.46 | 0.38 | 10.8 | 4.7 | 3.6 |

Root mean squared error (RMSE) on the test set, training time, and inference time per forecast on the three datasets. Zeros occur due to rounding.

### Table 2 | Comparison of XAI methods

| XAI method | SHAP values | Runtime/explanation | | | Feature importance error | Local explanation error | Instability | |
|---|---|---|---|---|---|---|---|---|
| | | Synthetic [s] | Load [s] | Price [s] | [%] | | Load | Price |
| TFT | no | **0.01** | **0.03** | **0.01** | 8.04 | – | 5.57 | 5.39 |
| KernelSHAP | approx. | 252.03 | 136.65 | 190.35 | 11.09 | 0.063 | 0.50 | 0.83 |
| PermutationSHAP | approx. | 1124.16 | 484.34 | 789.26 | 7.98 | 0.056 | 0.75 | 0.85 |
| WindowSHAP | approx. | 7.84 | 3.54 | 5.89 | 5.41 | 0.049 | **0.40** | **0.73** |
| SHAPformer | exact | 21.90 | 0.60 | 0.73 | **1.64** | **0.016** | 1.60 | 1.82 |

The feature importance error and local explanation error are evaluated on the synthetic data with ground-truth explanations, and the feature importance instability is defined as the standard deviation of the feature importance values during across experiment repetitions.

addition, SHAPformer leverages attention manipulation to exclude absent feature groups, eliminating the need for sampling and repeated model evaluations, which makes the evaluation of all coalitions computationally feasible while enabling an additional speedup. However, SHAPformer's inference speed advantage comes at the cost of increased training time, which is between two and thirteen times longer than that of TFT or the standard Transformer. Inference times of both the Transformer and SHAPformer are higher on the synthetic dataset than on the empirical datasets, as the models are larger (see the Supplementary Results for the hyperparameters) and the synthetic dataset includes one additional covariate. SHAPformer is faster than WindowSHAP on the load and price datasets with 13 features each, but is slower on the synthetic dataset with 14 features due to the exponential growth in the number of evaluated coalitions in SHAPformer.

TFT is the fastest explanation method because it produces explanations with a single model evaluation. However, it is not directly comparable to the SHAP-based approaches, as it provides only feature importance values but does not indicate how individual features influence the predictions, making it less informative than SHAP-based approaches.

SHAPformer's forecast accuracy is comparable to or better than that of the other models reported in Table 1. All evaluated models outperform the persistence baseline, which simply predicts the value from one week earlier. The Transformer achieves the lowest forecast error on the synthetic and load datasets, with SHAPformer closely following, exhibiting a forecast error that is only about 1% higher. On the price dataset, SHAPformer is the most accurate model, outperforming TFT and the Transformer by 4.4% and 7.9%, respectively. On the empirical datasets, SHAPformer outperforms Linear Regression, XGBoost, and TFT. The higher forecast error of TFT on the load dataset stems from (a) a larger variance across experiment repetitions (although SHAPformer and the Transformer outperform TFT in all runs), and (b) large forecast errors before Christmas, where TFT's causal attention mechanism prevents the model from using the holiday feature of the later days when predicting the first days in the forecast horizon.

## Successful validation of SHAPformer's explanations on synthetic data

Having demonstrated SHAPformer's forecast accuracy and fast inference, we validate its explanations using a synthetic dataset with known ground-truth explanations. The synthetic time series incorporate daily, weekly, and annual seasonality, along with dependencies on covariates. Ground-truth explanations are obtained by applying SHAP to the data generation process. Details of the dataset and ground truth generation are provided in Synthetic dataset.

SHAPformer's global feature importance is close to the ground truth, except that it underestimates the importance of the month feature (Fig. 2A). In contrast, PermutationSHAP and WindowSHAP deviate from the ground truth, particularly for key features such as the past target (called "load"), the hour of day, and the day of the week. TFT is more difficult to interpret, as it produces separate sets of feature importance values for past and future features. Nonetheless, it is observable that neither of the sets resembles the ground truth, nor does a linear combination of the two. Notably, both SHAPformer and PermutationSHAP effectively filter out irrelevant features such as the two noise covariates, while the other methods assign them importance values greater than zero, although the target variable is independent of the noise covariates. We calculate the feature importance error, defined as the mean absolute error of the feature importance values shown in Fig. 2A with respect to the ground truth, and report it in Table 2. We find that SHAPformer closely matches the ground truth feature importance, deviating only by 1.64 percentage points, whereas the feature importance error of the second-best WindowSHAP is more than three times larger. Since ground-truth explanations are available

for all samples in the synthetic dataset, they can also be used to evaluate the local explanations of the SHAP methods. Table 2 reports the local explanation error, defined as the mean absolute error of the local explanations averaged across all samples, forecast horizons and feature groups. Again, we find that SHAPformer best matches the ground truth, resulting in a three- to four-times lower local explanation error than the other SHAP algorithms.

Next, we compare SHAPformer's learned feature dependencies (Fig. 2B) with the ground truth (Fig. 2C). A positive SHAP value indicates that a feature increases the model's prediction relative to the mean prediction, whereas a negative SHAP value indicates a decrease. The dot color indicates an interacting variable. In general, SHAPformer's feature dependencies align well with the ground truth, except that it underestimates the importance of the month. In panel (a) of Fig. 2B, C, the previous week's load has a linear effect on the prediction. A given past load results in a higher SHAP value at night (bright dots) than during the day (dark dots), because the load is expected to be lower at night. In panel (b), the dependence on the hour of day forms a half-sine wave, which is exactly the daily pattern used during data generation. This interacts with the multiplier: negative multipliers (dark blue dots) decrease the effect of the hour of day, whereas positive multipliers (light green to yellow) increase its effect. Panel (c) reflects the multiplicative shrinking effect used in the data generation to model weekends (days five and six), with SHAP values closer to zero than on weekdays – negative at night and positive during the day. Panel (d) shows that holidays shrink the predicted load by increasing negative past loads and reducing positive ones, while non-holidays exhibit a reversed but less pronounced effect. In panel (e), the month feature has a sinusoidal effect in the ground truth, which SHAPformer partially captures, though with a reduced amplitude. Finally, panel (f) shows an X-shaped pattern for the multiplier feature: higher multipliers amplify positive previous loads and reduce negative ones, whereas lower multipliers have the opposite effect.

Overall, the agreement of SHAPformer's feature importance and feature dependencies with the ground truth suggests that the model captures the underlying dependencies in the data. In addition to the global explanations, SHAPformer's local explanations are also consistent with the ground truth, which is shown in Supplementary Results. The dependence plots and local explanations created with PermutationSHAP and WindowSHAP are presented in the Supplementary Results for comparison.

## Insights into electrical load and electricity price forecasts

With SHAPformer validated on synthetic data, we examine its global explanations for the last twelve months of the load and price datasets, which were excluded from model training. In these cases, no ground truth is available, so we use domain knowledge to assess the quality of the explanations.

On the electrical load dataset analyzed in Fig. 3, SHAPformer identifies the past load as the most important predictor, followed by the day of the week and the hour of the day. Features such as month, temperature, and holidays contribute less and show similar levels of importance, while precipitation has almost no importance. The explanations of PermutationSHAP and WindowSHAP are similar to each other but differ from SHAPformer's explanations, emphasizing the hour of day and the day of the week, and assigning much lower importance to the past load than SHAPformer. For TFT, the interpretation is not straightforward, as it outputs two distinct sets of feature importance – one for past inputs and one for future covariates – with an unclear relationship between them. In TFT's case, the most important past feature is the load, while the hour of day, temperature, and day of the week are the most important future features.

Figure 3B displays SHAPformer's learned feature dependencies. SHAP values are expressed in terms of standardized electrical load, where a value of 1.0 corresponds to one standard deviation

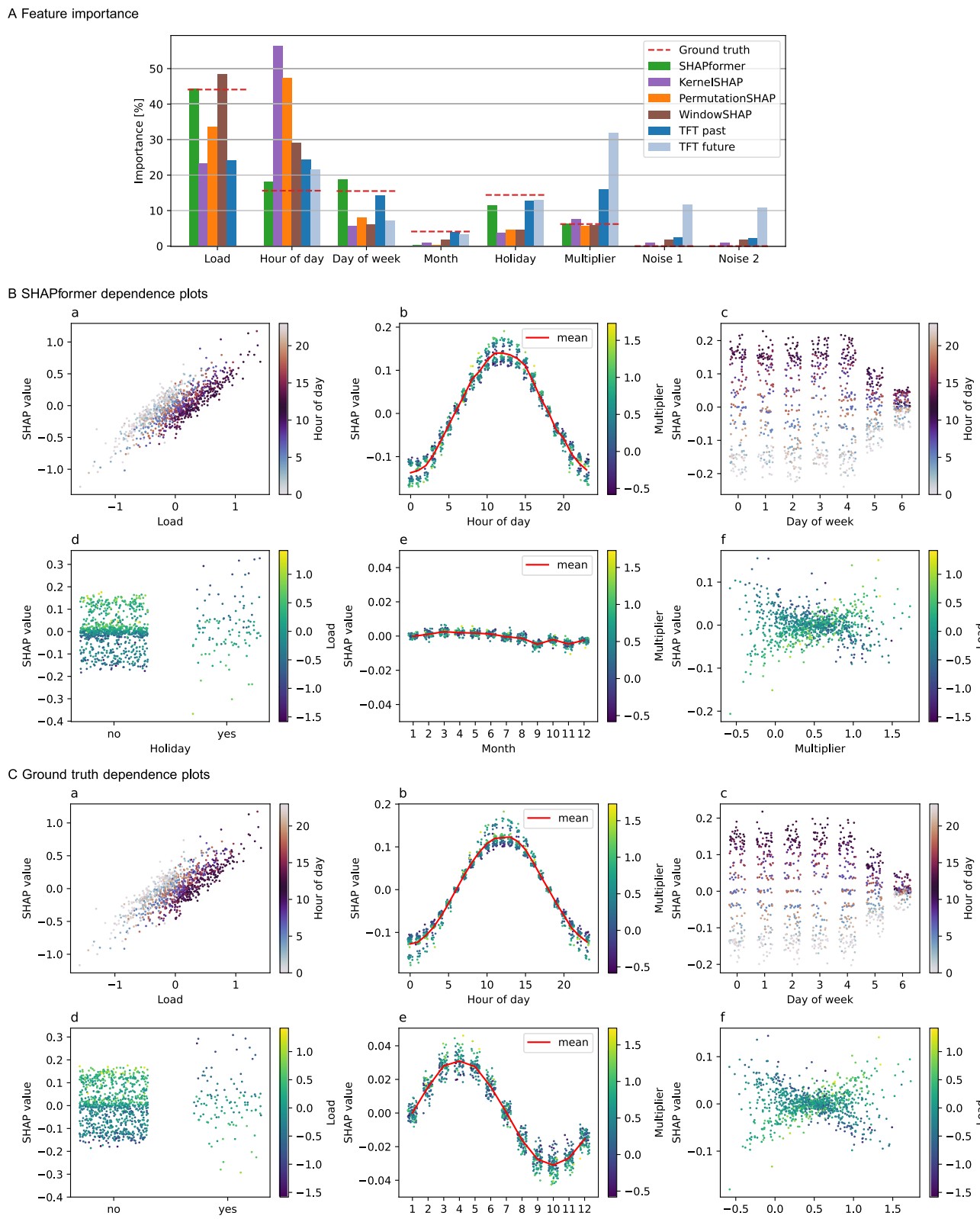

**Fig. 2 | Global explanations on the synthetic test data. A** SHAPformer approximates the ground-truth feature importance well (TFT: Temporal Fusion Transformer). **B**, **C** Dependence on the six most important features. Dot color indicates an interacting variable. For discrete variables, noise was added in the x-direction for visibility reasons.

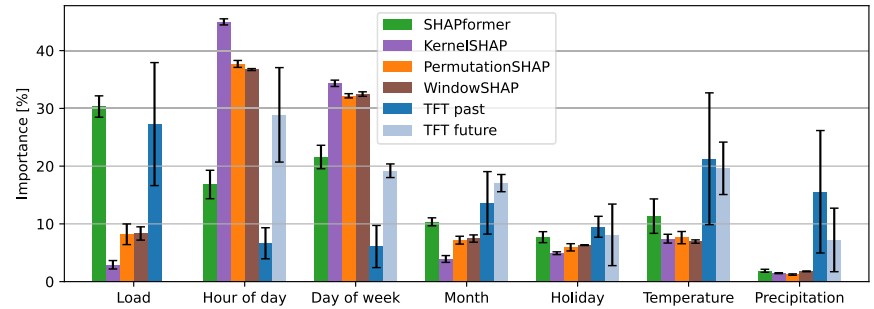

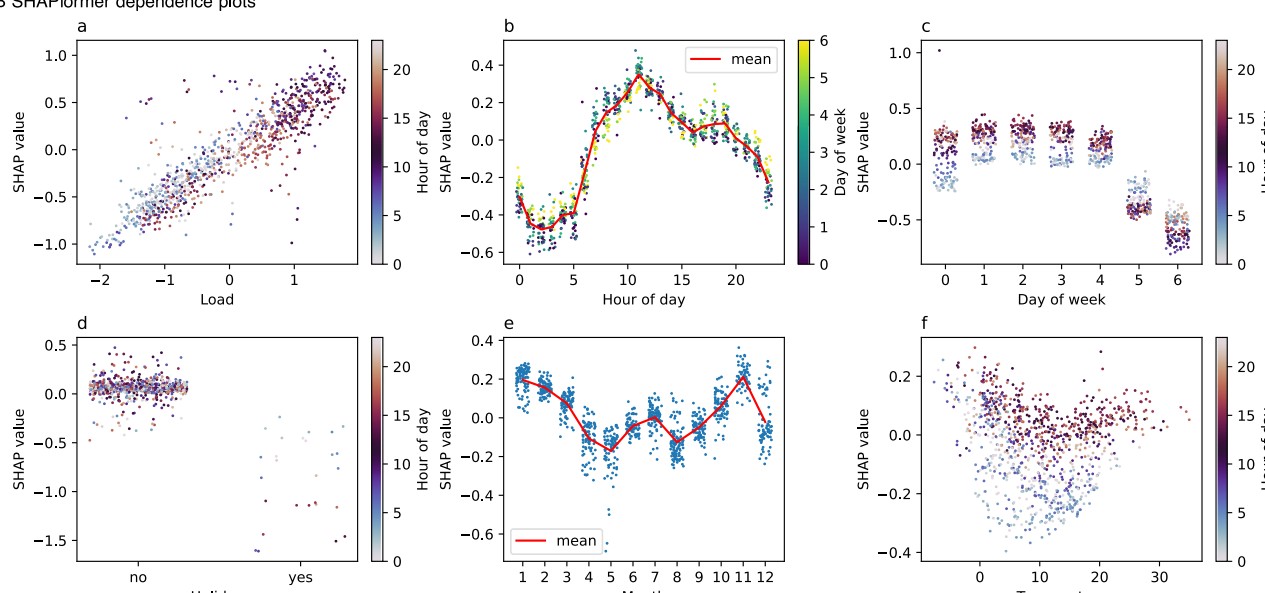

**Fig. 3 | Global explanations on electrical load data from TransnetBW. A** Feature importance scores by SHAPformer, a Transformer model explained using different SHAP algorithms, and Temporal Fusion Transformer (TFT). **B** Dependence plots from SHAPformer, with feature values on the x-axis and corresponding SHAP values on the y-axis. Dot color indicates an interacting variable. For discrete variables, noise was added in the *x*-direction for visibility reasons.

(~ 1550 MW). In panel (a), a linear relationship between the previous week's load and the forecasted load is observed. Notably, SHAP values for a given load value are higher at night (bright dots) – likely because a load considered low during the day can be unusually high at night. Panel (b) shows the typical daily load pattern, with lower values at night and two peaks around noon and in the evening. On Sundays (yellow dots), SHAP values tend to be closer to zero. Panel (c) illustrates that the predicted load is lower on Saturdays and Sundays, especially during the day (dark dots). Holidays, shown in panel (d), consistently reduce the predicted load, although the effect is weaker at night (bright dots). In panel (e), the month feature reflects a seasonal pattern with higher loads in winter, except in December, likely due to reduced industrial activity during the Christmas period. Finally, panel (f) shows that temperature influences the predicted load on cold days, with particularly strong increases when daytime temperatures are below 15 °C or nighttime temperatures are below 0 °C.

Figure 4 presents local SHAPformer explanations for two forecasts in December. In both cases, the hour-of-day feature lowers the predictions at night and shows two daily peaks – mirroring the patterns observed in the dependence plots. The day-of-week feature reduces the predictions on weekends, as expected. Interestingly, in the left-hand example, the month feature has a positive effect despite being in December – likely because the previous week falls in November, which is represented in the month feature of the encoder input. In contrast, the right-hand example, which is later in December,

shows a strongly negative month effect. The holiday feature has a positive impact during weekdays, suggesting that the model associates December with lower loads in general due to the extended holiday season, except during weeks without holidays. In the left-hand example, the colder temperatures in the early forecast days increase the prediction, while the warmer temperatures in the last two days decrease it. In the right-hand example, the temperature is more stable and has a consistently positive effect on the prediction. The effect of the past load also differs between the two examples: in the left-hand example, the past load increases the prediction on weekdays, whereas in the right-hand example, it decreases the prediction from days 3 to 7 – especially influenced by the last two input days, which show a lower average load than the first days.

Overall, the observed dependence patterns align well with domain knowledge, reinforcing confidence in SHAPformer's predictions. The December pattern – where a lower load is treated as the default and non-holidays as exceptions – appears unexpected but can be plausibly explained by the extended holiday period during that time.

The feature importance and feature dependence plots for the price data (see Fig. 5) again align with our domain understanding[55]. Aside from the past price and temporal signals, wind speed is used as an important feature with approximately a linear negative dependency. Electricity prices are often assumed to increase linearly with residual load, i.e., load minus must-run capacity. As wind and solar power, entering our model via the irradiance feature, are typically

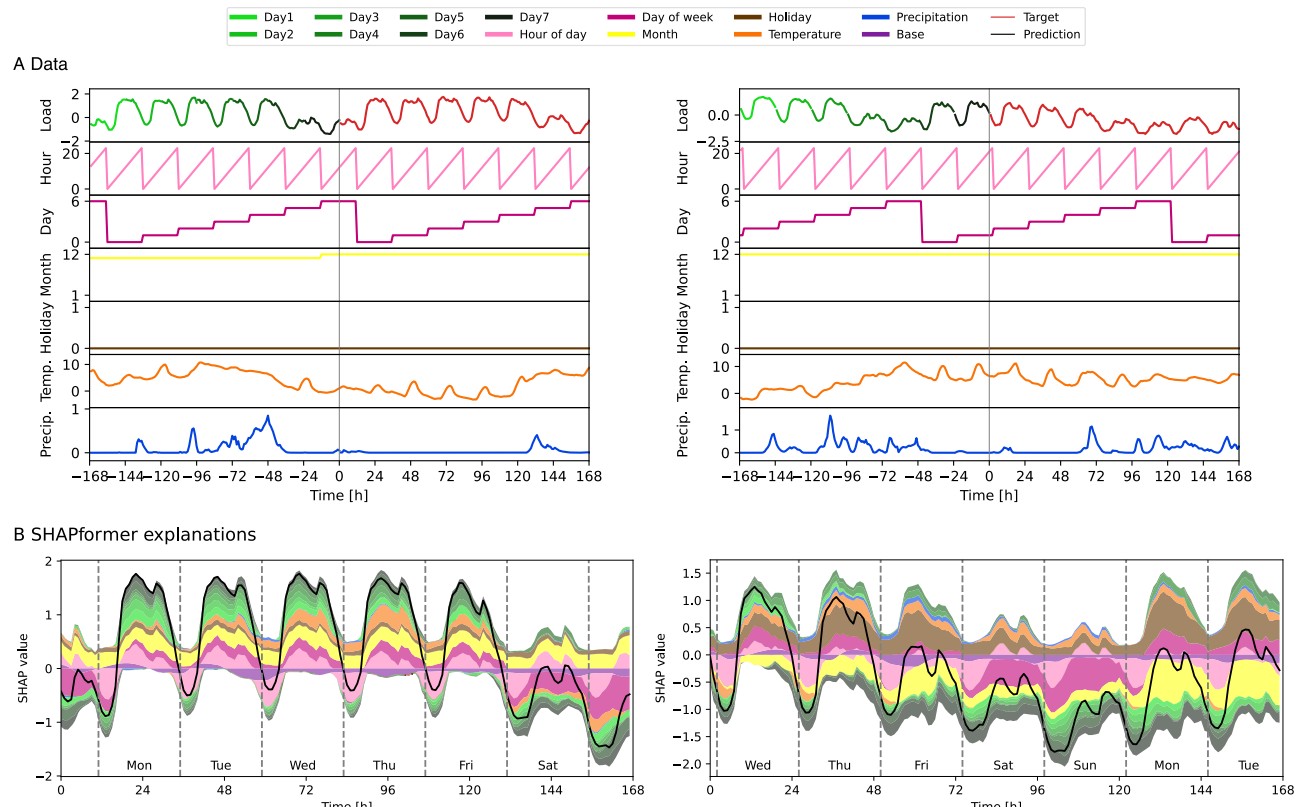

**Fig. 4 | Local explanations of examples from the real-world load data of TransnetBW. A** The input data and target for two examples from the electrical load dataset. The start of the forecast horizon on the left-hand side is December 1, 2019, 13:00 and on the right-hand side, December 17, 2019, 22:00. **B** The corresponding explanations by SHAPformer.

considered must-run generation, they will reduce the price, as we see in our model. The hour-of-day feature resembles the typical price peaks in the morning and the evening, while the day-of-week and holiday features reduce the price primarily on non-working days. The month feature shows no clear pattern; presumably, the model reflects high prices associated with specific periods in the training data.

## Discussion

We introduced SHAPformer, an accurate and efficient approach for explainable time-series forecasting. SHAPformer generates SHAP explanations for Transformer-based models several orders of magnitude faster than the established PermutationSHAP. This significant speedup results from two key mechanisms: grouping input features to reduce the number of coalitions and manipulating the attention weights to estimate marginal contributions without sampling. Both mechanisms contribute to reduced runtime. WindowSHAP, which applies feature grouping but not attention manipulation, is faster than PermutationSHAP. However, only SHAPformer achieves inference times below one second per explanation.

SHAPformer has been successfully validated on synthetic data, where it closely reproduces the ground truth in terms of feature importance, dependence patterns, and feature interactions. It correctly identifies irrelevant noise features as non-informative and captures key relationships, including daily patterns, multiplicative effects of holidays and weekends, and temperature interactions. The only exception is the month feature, whose importance SHAPformer appears to underestimate, likely because its influence in the data generation process is small and comparable to the noise in the target time series (see Synthetic dataset).

SHAPformer combines the predictive strength of Transformers with an efficient method for computing calibrated SHAP explanations.

However, training on masked inputs to enable exact attribution can slightly reduce forecasting accuracy compared to a standard Transformer. In our experiments, this results in a small increase in RMSE on two datasets, while SHAPformer performs better on the electricity price dataset. We consider these differences acceptable given the substantial gains in interpretability and efficiency: SHAPformer produces explanations that are more closely aligned with ground truth SHAP explanations and reduces the computational cost of explanation generation by up to three orders of magnitude. Thus, SHAPformer represents a trade-off between marginally reduced predictive accuracy and improved explainability and runtime.

Existing SHAP algorithms are either true to the model or true to the data[56]. Conditional sampling tends to produce explanations that are true to the data, while off-the-manifold sampling generates explanations that better reflect the model's internal behavior[40]. We argue that SHAPformer achieves both: it is true to the model and true to the data. This is made possible by its sampling-free design and training strategy, which builds robustness to absent features. In the presence of correlated features, SHAPformer learns to predict accurately using any of the correlated inputs, distributing the predictive contribution among them. This contrasts with models trained on the full feature set, which may rely on arbitrary feature combinations, leading to explanations that misrepresent underlying data relationships. SHAPformer's strong alignment with the ground-truth explanations on synthetic data supports this claim.

Applying SHAPformer to real-world load data provides valuable insights into how different features influence the model's predictions. The most important predictor is the load from the previous week, followed by the day of the week and the hour of the day. Other features – such as month, temperature, and holidays – also contribute, while precipitation has minimal impact. This contrasts with the Transformer

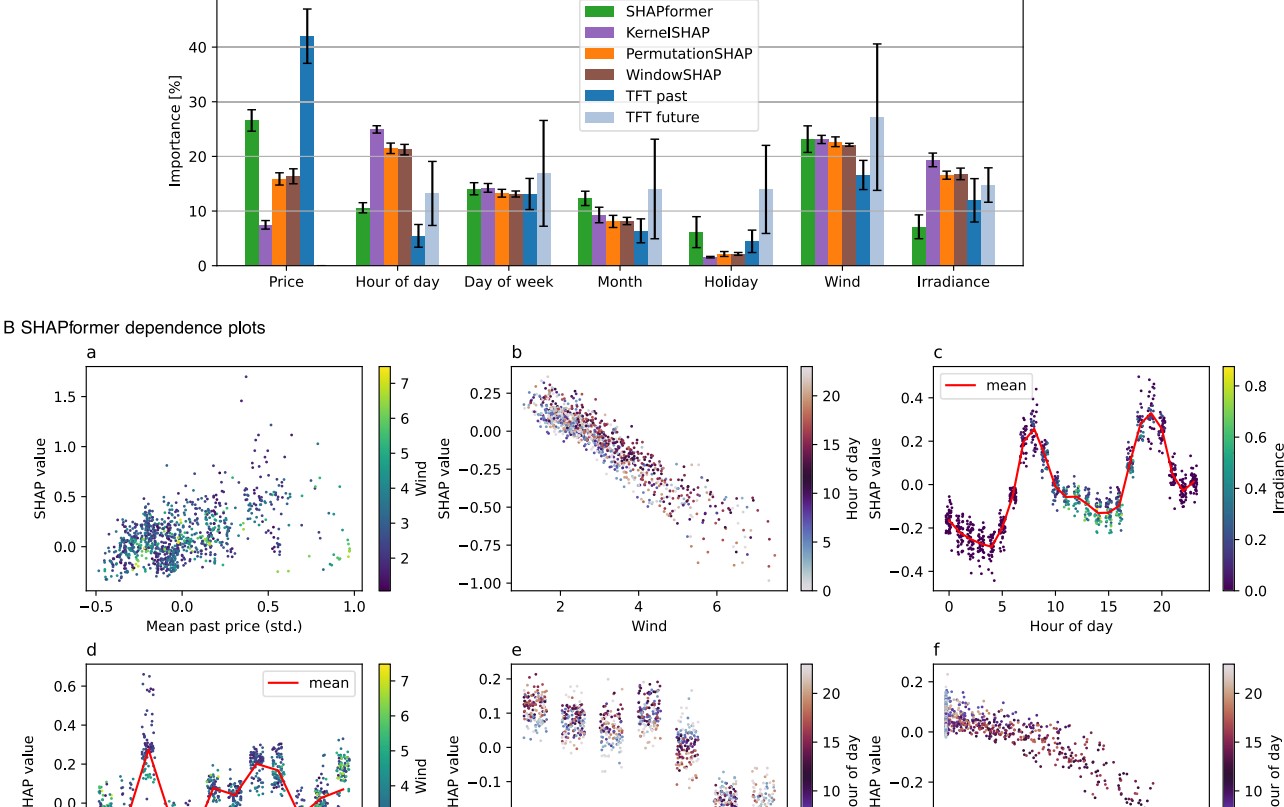

**Fig. 5 | Global explanations on electricity price data. A** Feature importance scores by SHAPformer, a Transformer model explained using different SHAP algorithms, and Temporal Fusion Transformer (TFT). **B** Dependence plots from SHAPformer, with feature values on the x-axis and corresponding SHAP values on the y-axis. Dot color indicates an interacting variable. For discrete variables, noise was added in the x-direction for visibility reasons.

model explained by PermutationSHAP and WindowSHAP, which emphasizes temporal features more heavily and downplays the role of the past load. This difference highlights how global explanations can help differentiate between models with similar forecast accuracy but different learned dependencies. In our view, a forecasting model can reasonably rely on both calendar-based inputs or lagged load values to estimate typical load patterns. SHAP values are instrumental in uncovering which of these cues the model relies on. Given that SHAPformer aligns more closely with the ground-truth explanations on synthetic data, we are confident that its explanations on real-world data are likewise more faithful to the underlying data relationships than those from the comparison methods.

Despite its advantages, SHAPformer has several limitations. First, SHAPformer's fast inference comes at the cost of increased training time. Training with masked inputs requires the model to learn a more complex structure across varying feature subsets, and since it only sees partial inputs in each iteration, the effective amount of training data per epoch is reduced. Nevertheless, for applications where many forecasts need to be explained, the reduced inference time justifies the additional training effort. Second, the computational complexity of SHAPformer scales exponentially with the number of feature groups, because the model is evaluated for all $2^N$ subsets of $N$ feature groups. In our setup with seven input days and six or seven covariates (depending on the dataset), this full enumeration is computationally feasible and desirable, as it yields the most accurate SHAP estimates. However, in

scenarios with more feature groups, such as longer input sequences or additional covariates, this exhaustive approach may become impractical. Future work can address this issue by estimating approximate SHAP values based on a subset of the feature groups. Third, feature importance and local explanations can vary across training runs, as reflected by the instability metric in Table 2. This metric measures the standard deviation of the feature importance values over five model training and evaluation runs, and the results show that SHAPformer's feature importance deviates by one to two percentage points on average. Fourth, the empirical evaluation is limited to one synthetic dataset and two real-world datasets, since each dataset requires domain understanding to verify explanations, and we leave the evaluation on more datasets to future work. Finally, the synthetic benchmark relies on several assumptions: the ground-truth explanations are defined using SHAP values derived from the data-generating process, they assume that the learned model faithfully represents this process, and they respect dependencies between features (e.g., the hour-of-day can be inferred from the day-of-week feature, which indicates day changeovers). While these assumptions allow controlled evaluation, faithful explanations for models that do not follow these assumptions will differ from the ground truth.

In the future, the efficient SHAP calculation of SHAPformer can be adopted for other models than Transformers, by training them on feature subsets, which allows to run them on feature subsets at inference time. For models that do not rely on attention, feature dropout

could be used instead of attention manipulation. Training models in this way not only facilitates explainability but may also improve robustness to known anomalies and missing values. By dropping out these values with attention manipulation or equivalent mechanisms, such models may adapt more effectively to real-world data imperfections.

We release SHAPformer as a Python package[57] to make it available to researchers and practitioners. This will facilitate its application and evaluation across a range of domains beyond electrical load forecasting. SHAPformer can be applied in univariate settings – explaining the prediction based on windows of the past target alone – and in covariate-informed settings, thereby covering a variety of use cases. Given its successful validation on synthetic data and evaluation on two forecasting tasks, we are confident that SHAPformer will generalize well to other use cases.

We publish the synthetic dataset and the ground-truth explanations[58], so that they can be used for future evaluations of XAI methods. However, note that this ground truth is created on the basis of multiple assumptions, such as that the ground truth should resemble SHAP values and it should respect dependencies between features (e.g., the hour of day can be inferred from the jumps in the day-of-week feature). Other explanations might be valid under different assumptions, especially when the goal is to create explanations that are true to a model that has learned different dependencies. We view this evaluation approach as complementary to existing explanation metrics[59] and believe that it can be extended to other data modalities in the future.

## Methods

This section introduces SHAP and three algorithms to estimate SHAP values (PermutationSHAP, KernelSHAP and WindowSHAP) in Shapley Additive Explanations (SHAP). Next, our method SHAPformer is described in SHAPformer. Then, the synthetic dataset and ground truth generation processes are described in Synthetic dataset. The compared forecasting models are described in Baselines. Finally, the evaluation metrics are described in Evaluation metrics.

### Shapley additive explanations (SHAP)

**Shapley values.** Shapley values are a concept from game theory that is used to compute a player's contribution to the outcome of a cooperative game[17]. The player's contribution is defined as how much the game outcome changes on average when the player enters the game, considering all permutations of players entering the game. The Shapley value definition is the unique solution for the computation of players' contributions that fulfill the efficiency axiom (the sum of the contributions adds up to the outcome), the symmetry axiom (two identical players receive equal contributions), the dummy axiom (a player that does not contribute to any coalition gets a contribution of zero) and the additivity axiom (the contribution to the sum of two games equals the sum of the contributions of the two games)[16,60].

**Explaining machine learning models with SHAP.** The concept of Shapley values is used in machine learning to compute the contribution of features to the prediction of a model[16,60]. For this, the features are the players, and the model prediction is the game outcome. The exact Shapley values usually cannot be computed for computational reasons, and because it is not possible to integrate over the unknown distributions of absent features. Therefore, the Shapley values are estimated in practice and are then called SHAP values, in order to distinguish them from the exact Shapley values. The naive algorithm to compute SHAP values works as follows: For a set of features $\{v_1, \ldots, v_n\}$, consider all permutations $p_j, 1 \leq j \leq n!$. For a permutation $p_j$, go through all features in the order of the permutation. For a feature $v_i$, let $S$ be the features that precede $v_i$ in $p_j$. Then, the marginal contribution of $v_i$ is

computed as:

$$\phi(v_i, p_j) = f(S \cup \{v_i\}) - f(S) \qquad (1)$$

where $f(X)$ is the model prediction using a set of features $X$. The marginal contribution is the effect on the model prediction when adding $v_i$ to the feature set. The SHAP value for $v_i$ is defined as the average marginal contribution of $v_i$ in all permutations. It can be computed more efficiently by iterating over all $2^n$ feature subsets instead of all $n!$ feature permutations:

$$\text{SHAP}(v_i) = \sum_{S \subseteq V \setminus \{v_i\}} \frac{(n - 1 - |S|)! \cdot |S|!}{n!} \cdot (f(S \cup \{v_i\}) - f(S)) \qquad (2)$$

The first part of the formula is a weight, defined by the fraction of permutations in which $v_i$ gets added to $S$, and the second part is the marginal contribution of $v_i$, defined as the difference of the model prediction with and without $v_i$. Note that to compute all feature contributions, it is necessary to compute the model predictions for all $2^n$ feature subsets $S \subseteq V$ (also called coalitions)[16,60].

Usually, a machine learning model trained on a set of features requires all features for inference, which means that it does not allow for predictions based on feature subsets. Therefore, a common approach to estimate predictions based on feature subsets is to marginalize out the effect of the absent features via a Monte Carlo integration – that is, by sampling their values $k$ times from a background dataset (e.g., the training data). Consequently, the model is called $k$ times for each of the feature subsets.

**PermutationSHAP.** PermutationSHAP is a more efficient way of estimating SHAP values by calculating marginal contributions only for a subset of the coalitions[60]. It samples $K$ random feature permutations. For each permutation $p_j$, it starts with the empty feature set and iteratively adds features in the order of the permutation. The marginal contribution of each feature is computed as the prediction with that feature minus the prediction without that feature, i.e., $f(S \cup \{v_i\}) - f(S)$ as above, where $S$ is the set of features preceding $v_i$ in $p_j$. As in the naive algorithm, predictions based on feature subsets are estimated by a Monte Carlo integration over the absent features (i.e., by sampling their values from background data). Each permutation is used twice, where features are added once in the forward and once in the backward direction. The SHAP value estimate is then computed as the average marginal contribution of a feature over all sampled permutations.

PermutationSHAP is commonly used in practice, but it has two drawbacks:

1. It is costly to compute for large feature sets. In our forecasting setup on the electrical load dataset with $168 \times 7$ past features and $168 \times 6$ future features (2 184 features in total), ten feature permutations (each run forward and backward), and 100 samples to estimate distributions of absent features, $2\,184 \times 10 \times 2 \times 100 = 4\,368\,000$ model calls are needed to compute a single local explanation of a forecast. The runtimes reported in Table 1 are measured with ten permutations. As this is not feasible to evaluate for many samples, we instead reduce the number of permutations to two to generate the local and global explanations shown in the figures.
2. When feature dependencies are not considered during Monte Carlo sampling, unrealistic samples are created, which are out of the distribution of the training data and can potentially lead to arbitrary model behavior. For example, when calendar features are sampled independently, it can happen that Monday midnight is followed by Friday noon. Other examples of unrealistic counterfactuals are 30 °C outside temperature in winter, quick jumps in

the electrical load or temperature values, high temperatures or loads at night, and many more.

Note that many algorithms exist that estimate SHAP values[39] and not all of them require sampling. For text and images, usually no sampling is required. Instead, patches of images are blurred or replaced by unicolored pixels. Words are replaced by "..." or removed entirely. Similarly, it is possible to replace time-series features by a baseline value of zero, their negative values or the values in reversed order, as is done in perturbation methods[61]. However, choosing a baseline value is not straightforward, and the chosen baseline affects the explanations. After standardization of the continuous time-series features, a value of zero represents the time series' mean, so applying a zero baseline would only explain the difference to the mean. When a non-holiday is chosen as the baseline for the holiday feature, non-holidays will be attributed SHAP values of zero, as the feature value and the baseline value are the same. For the day-of-week feature, no meaningful sequence of weekdays can be chosen as a baseline, as every sequence appears equally often, whereas constant values (e.g., seven consecutive Mondays or 168 times midnight) are unrealistic.

**KernelSHAP.** In addition to PermutationSHAP, we also evaluate KernelSHAP[16]. KernelSHAP is a model-agnostic method that estimates SHAP values by approximating the Shapley value formulation with a weighted linear regression. To this end, the model is evaluated on perturbed versions of the input, where subsets of features are replaced with values sampled from background data to simulate their absence. The resulting model outputs are then used to fit a linear surrogate model whose coefficients correspond to the estimated SHAP values, with weights chosen according to the Shapley kernel to ensure consistency with the Shapley value formulation. Because KernelSHAP is model-agnostic, it can be applied to arbitrary forecasting models in the same way as PermutationSHAP, allowing us to use it as an alternative baseline for comparison.

**WindowSHAP.** We develop a custom masker to mitigate the sampling and efficiency issues of PermutationSHAP. The custom masker is a variant of WindowSHAP[34] with seven windows for the past target and one window for each exogenous variable. Given an example for which an explanation is to be computed, and a feature coalition (i.e., a subset of the features), the custom masker defines how the features absent in the coalition get replaced in order to generate alternative samples, which are used in the Monte Carlo estimation.

First, we reduce the number of features by defining groups of features that get altered together. The past load is split into seven groups of 24 features, representing the past seven days. All features belonging to the same exogenous time series form one feature group. This makes 13 feature groups in total for the real-world data, and 14 feature groups in total for the synthetic data, which exhibits one exogenous feature more. As a result, the model is called $13 \times 10 \times 2 \times 100 = 26\,000$ times to create a local explanation of a forecast on the real-world data, using 10 permutations and 100 background samples.

The custom masker samples each feature group differently, respecting the characteristics of the feature and creating more realistic samples than PermutationSHAP. For a load feature group, 24 consecutive values from the training data are sampled, so that these 24 values form a smooth and realistic curve in themselves (but not necessarily realistic in the context of the other load values and exogenous features). The month values are increased by a constant, randomly chosen offset between 0 and 11 months. In the same way, the day-of-week and hour-of-day values are offset by 0 to 6 days, respectively 0 to 23 hours. For the temperature feature, the entire temperature curve is replaced by 336 consecutive values from the training data.

**From local to global explanations.** For all SHAP algorithms, the local explanations of $m$ samples are aggregated to feature importance values by summing up the absolute SHAP values of a feature group in all samples, and then calculating percentages on the summed values. We set $m = 300$ for the slow PermutationSHAP and KernelSHAP, and $m = 1000$ for the other SHAP algorithms.

Another form of global model explanation are the dependence plots shown in Figs. 2 and 3. For these, $m$ samples result in $m$ dots, each with the feature value on the x-axis and the SHAP value on the y-axis. One explanation results in 168 SHAP values for the 168 predicted time steps, but only the SHAP values for the 24-hour forecast (i.e., the 24th value in the forecast horizon) are shown in the dependence plots. Similarly, there are multiple feature values that could be shown on the x-axis (168 for the past load and 336 for each of the covariates), of which we chose the load of one week before the predicted time step for the load plot, and the future value of the predicted time step for all other plots.

## SHAPformer

**Attention manipulation.** As the main building block of the Transformer[53], the attention mechanism[62] computes a context-dependent embedding of a query vector given an arbitrary number of equally-dimensioned key vectors as context. More formally, there is a query vector $q$ and key vectors $k_1$ to $k_n$. With each key vector $k_i$ there is a value vector $u_i$ associated. The attention score $a_i$ of a key vector $k_i$ is computed as

$$a_i = \frac{q^T k_i}{\sqrt{d}}, \qquad (3)$$

where $d$ is the vector dimension. The attention scores $a_1$ to $a_n$ are then soft-maxed in order to compute attention weights $\alpha_1$ to $\alpha_n$ which sum up to one:

$$\alpha_i = \frac{\exp(a_i)}{\sum_{j=1}^{n} \exp(a_j)} \qquad (4)$$

The output $o$ of the attention mechanism is finally computed as

$$o = \sum_{i=1}^{n} \alpha_i \cdot u_i. \qquad (5)$$

Attention manipulation[49] allows dropping out individual vectors $k_i$ by setting their attention score to $a_i = -\infty$, so that the attention weight $\alpha_i$ becomes zero after the softmax operation. This means that the weight of vector $u_i$ in the weighted sum is set to zero, and therefore no information from $u_i$ is contained in the attention output, and the following layers of the model have no access to $u_i$.

In the following, we make use of attention manipulation to restrict the model's access to feature groups that are absent in a coalition.

**SHAPformer architecture.** Figure 6A shows the SHAPformer architecture. The model has a lookback size of 168 and a forecast horizon of 168. The 168 past feature vectors are given to the encoder. Each past vector contains the historic value of the target variable and exogenous features. The 168 future feature vectors are fed to the decoder. Each future vector contains the exogenous features.

SHAPformer uses a masked feature-attention mechanism to compute time step embeddings, as shown in Fig. 6B. This mechanism computes an embedding of the feature values belonging to one time step. For each variable, the model maintains a learned embedding. For categorical variables with $k$ possible values, $k$ randomly initialized embedding vectors are used. For continuous variables, a linear layer is used to embed the feature values. A query vector is computed as a weighted sum of the embedded variables, where dropped-out features

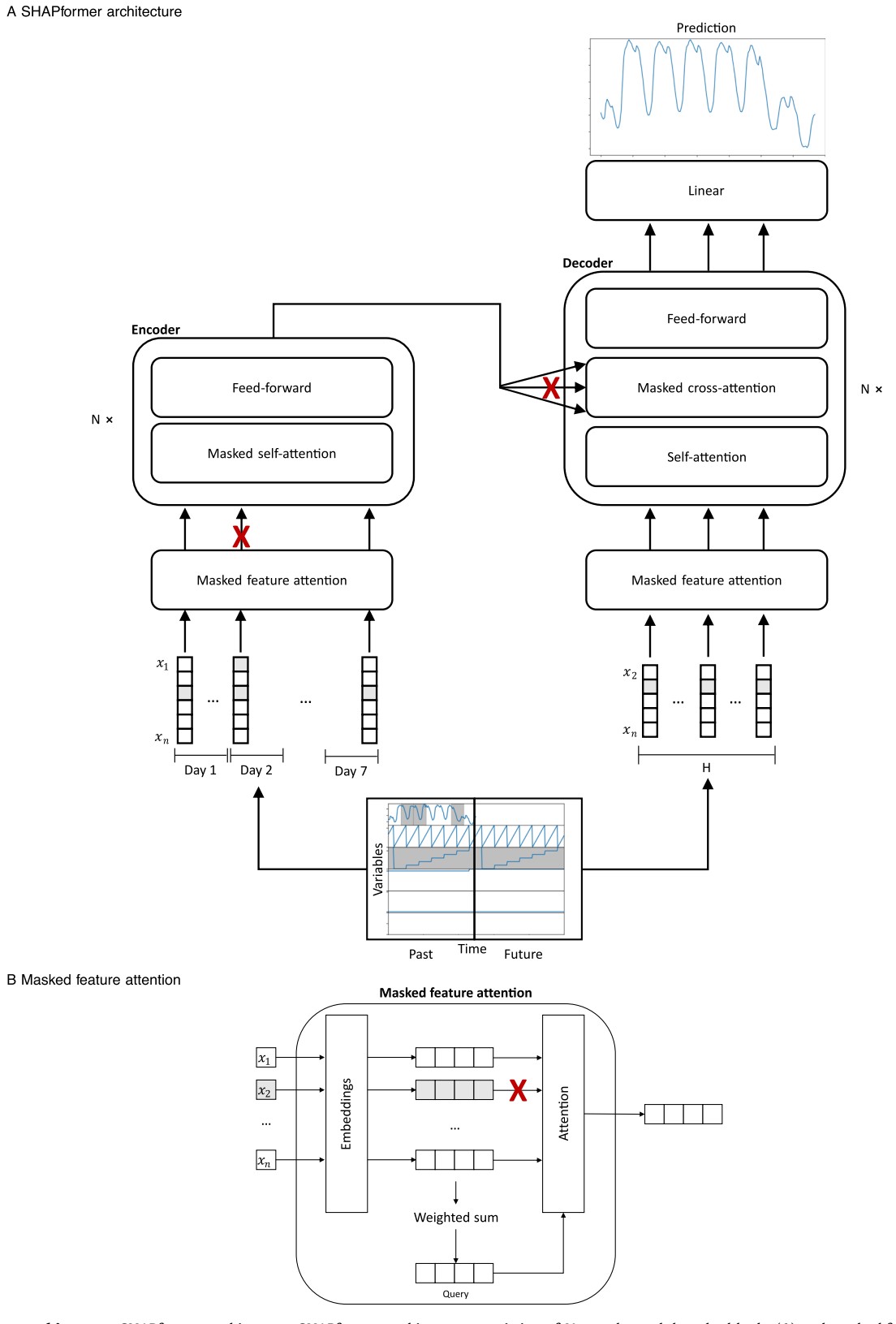

**Fig. 6 | SHAPformer architecture.** SHAPformer architecture. SHAPformer architecture consisting of *N* encoder and decoder blocks (**A**) and masked feature attention (**B**). Masked feature attention drops out variables. Masked self-attention and masked cross-attention drop out time steps.

get zero weight, and all others weight one. Then, attention is used to create the final embedding, based on the query vector and the embedded feature values as key vectors. In the masked feature-attention mechanism, the attention weight of absent features is set to zero, so that the model has no access to their embeddings ($v_2$ is masked in the example in Fig. 6B). With this approach, SHAPformer can be run on subsets of the exogenous features. A positional encoding is added to the embedding vectors to maintain their order.

For the load feature, masked self-attention in the encoder and masked cross-attention in the decoder are used to restrict access to 24 time steps belonging to the same load feature group.

The encoder gets 168 embeddings for the 168 past time steps as input. Multiple Transformer encoder layers are stacked, each consisting of self-attention and linear layers – both equipped with add and norm mechanisms (not shown in the figure). The self-attention in the encoder operates within the past time steps. The self-attention is masked, i.e., the attention weight for absent time steps is set to zero, so that they are not used to calculate the attention output.

The decoder gets 168 vectors for the 168 future time steps as input. It consists of multiple Transformer decoder layers using self-attention and cross-attention to access the encoder output. The self-attention in the decoder operates within the future time steps, whereas the cross-attention fuses information from the encoder output into the decoder representation of future time steps. The cross-attention is masked, i.e., the attention weight for absent time steps is set to zero, so that the decoder has no access to them (days 2, 3 and 6 in the example in Fig. 6A). Finally, a linear layer is used as prediction head to transform the 168 output vectors of the decoder into 168 scalar values, which are the model prediction for the next 168 time steps.

During model training, random masks are sampled for every example that is given to the model, so that each exogenous feature group and each day has a 50% chance of being masked. Thereby, the model learns to create robust forecasts using subsets of the features. Without masked training, the model could potentially behave arbitrarily when a feature is absent at inference time.

**Owen values.** Owen values[63] are a variant of SHAP values for games with a coalitional structure. First, the contributions of the coalitions are computed, and then they are broken down further into contributions of the individual players. The formula is slightly different from the formula for SHAP values:

$$\text{Owen}(v_i) = \sum_{R \subseteq M \setminus \{k\}} \sum_{T \subseteq B_k \setminus \{v_i\}} \frac{1}{|M| \cdot |B_k|} \frac{1}{\binom{|M|-1}{|R|}} \frac{1}{\binom{|B_k|-1}{|T|}} (f(Q \cup T \cup \{v_i\}) - f(Q \cup T))$$

(6)

with $M = \{B_1, ... B_l\}$ being the set of coalitions, $B_k$ the coalition containing $v_i$, and $Q = \bigcup_{r \in R} B_r$ the union of features over a subset of coalitions $R$. We use the Owen value formula in SHAPformer in order to treat the past load equally as the exogenous features on the coalition level, and then break the past loads' SHAP value down further into SHAP values of the seven input days.

Note that in our case, when the load of a past day is absent, the other features of that day are also absent, as a result of the masked self-attention in the encoder and masked cross-attention in the decoder. We assume that this does not affect the Owen values, because the past day's features are used as context information for the past day's load – when the load is not available to the model, the effect of the past day's features on the forecast is negligible.

## Synthetic dataset
**Data generation.** We create a large synthetic time-series forecasting dataset in order to verify the explanations from SHAPformer. Since the data synthetic data generation process is known, it allows to evaluate the quality of the explanations returned from SHAPformer and other XAI methods.

The dataset consists of 120 000 samples. Each sample contains two weeks of hourly values from the target variable and multiple covariates. The first week of the target variable is used as input to the model, and the model has to predict the second week of the target variable. We call the target variable "load", as in our real dataset. For the covariates, both weeks are used as input to the model. That is, we assume a perfect forecast of the covariates for the next week.

The generation of an example follows a multi-step procedure:
- A month $\in [1, 12]$, start weekday $\in [0, 6]$ and start hour $\in [0, 23]$ are sampled with uniform probability.
- A base load is sampled uniformly in the range $(-0.5, 0.5)$.
- The month has an additive effect on the base load. A value of $0.1 \cdot \sin(\text{month} \cdot 2\pi/12)$ gets added to the base load.
- A daily load curve of 24 values is generated as $factor \cdot (\sin(h - 0.5 \cdot \pi) + \alpha \cdot \sin(h \cdot 2\pi/24) + \beta \cdot \cos(h \cdot 2\pi/24))$, with $factor \in (0.5, 1)$, $\alpha, \beta \in (-0.5, 0.5)$ uniformly sampled.
- A uniformly sampled pattern of 24 values with mean 0 and a standard deviation 0.1 is added to the daily load. We call the result the day_pattern and use it as the load curve for workdays.
- The Saturday pattern deviates from the workday pattern by a multiplicative factor and an additive deviation. It is generated as $s_1 \cdot \text{day\_pattern} + \text{normal}(0, 0.1)$, with $s_1 \in (0.5, 0.9)$ uniformly sampled, and $\text{normal}(0, 0.1)$ generating 24 values from a normal distribution with mean 0 and standard deviation 0.1.
- The Sunday pattern is $s_2 \cdot \text{day\_pattern} + \text{normal}(0, 0.1)$, with $s_2 \in (0.2, s_1)$.
- From the workday, Saturday and Sunday patterns, the 336 hourly values are produced by repeating the daily pattern, Saturday pattern and Sunday pattern, beginning from the start day of the week and start hour of the day. Each day has a 10% chance of being a holiday. On holidays, the Sunday pattern is always used, independent of the weekday.
- A temperature curve is generated by a random walk starting from a uniformly sampled value in $(-0.5, 0.5)$ and each step sampled from a normal distribution with mean 0 and standard deviation 0.02. The temperature has a multiplicative effect on the load. That is, the load is rescaled by the temperature as $\text{load} \cdot (0.5 + 0.5 \cdot \text{temperature})$.
- Finally, random noise is added to the load time series, sampled from a normal distribution with mean 0 and standard deviation 0.05.

The following covariates are used as inputs for the forecasting models: hour of day, day of week, month (all categorical), holiday (binary), temperature and two uncorrelated noise features (all continuous).

100,000 samples are used for training, 10,000 for validation and 10,000 for testing. Examples are shown in the Supplementary Results.

**Ground-truth explanations.** In order to validate the explanations from the different XAI methods presented in Successful validation of SHAPformer's explanations on synthetic data, we calculate ground truth SHAP values for the synthetic dataset. This is possible because we have access to the data generation procedure and can therefore explain the true data dependencies with SHAP. As SHAP is model agnostic, it can be used to explain an arbitrary function $f(x)$. Usually, in the context of XAI, $f$ is a machine learning model, but we set $f$ to the data generation process in order to calculate ground-truth explanations. The inputs $x$ to the data generation process consist of the past load, hour of day, day of week, month, holiday, multiplier and two noise features, each represented as a time series of 168 past and – for all inputs except for the load – 168 future values, and the output is the target load of the next week.

We compute ground-truth explanations for the examples in the test set of the synthetic dataset. To do so, we use PermutationSHAP on the data generation process. For a given test example and a permutation, the PermutationSHAP computes marginal contributions of the inputs based on subsets of the inputs. The inputs that are absent, i.e.,

not contained in the subset, are resampled 1000 times following the sampling process described above, and an alternative target load curve is generated using the data generation process and the resampled inputs. These 1000 alternative targets are then averaged, and the averaged values are used as the expected target load curve given the subset of the inputs. From the target load curves generated with different input subsets, marginal contributions and SHAP values are computed with the procedure described in Shapley Additive Explanations (SHAP).

While sampling alternative inputs, we make sure that no unrealistic combinations of inputs are generated. In particular, when an input $x_1$ depends on an input $x_2$ that is contained in the active subset, we do not resample $x_1$. The following dependencies are respected:

- If the day of the week is in the active subset, the hour of the day is not resampled (as it can be inferred from the day beginnings and endings).
- If the holiday feature is in the active subset and there is at least one holiday in the example, the hour of day is not resampled (for the same reason).
- If the load of the past week is in the active subset, all calendric information, as well as the multiplier of the last week, is not resampled (because they affect the load), as well as the load patterns for the workday, Saturday and Sunday (as changing any of these would affect the past load).

Note that the ground truth SHAP values are meant to be true to the data[56] under the assumptions stated above. However, a machine learning model could learn different patterns, so that explanations being true to the model instead of true to the data would deviate from our ground truth.

## Baselines
**Persistence baseline.** The persistence baseline predicts the value from one week before the predicted time step. It thereby respects daily and weekly seasonalities, but no dependencies on covariates. In its simplicity, the baseline is inherently interpretable.

**Linear regression.** The linear regression model is based on the following features: the load from 168 time steps before the predicted time step, the hour of day (sine and cosine encoded), the day of week (sine and cosine encoded), the month (sine and cosine encoded), whether it is a holiday (binary) and the temperature and precipitation values. As the model is linear, an explanation can be derived from the model coefficients.

**XGBoost Regressor.** The Extreme Gradient Boosting Regressor (XGBoost) uses the same features as the linear regression. The default hyperparameters, 100 estimators and a maximum depth of three are used. The model is able to return feature importance values, but no information is available on how the features impact the prediction.

**Time-series Transformer.** The time-series Transformer uses the same model architecture as SHAPformer, but it is trained without masking, so that the model always receives the full information about the last week and exogenous features for the next week. KernelSHAP, PermutationSHAP and WindowSHAP are used with the time-series Transformer to generate explanations.

**Temporal Fusion Transformer.** The Temporal Fusion Transformer (TFT)[46] is based on an LSTM encoder and an LSTM decoder[64], followed by a single Transformer[53] layer. It uses causal masking in the Transformer layer to prevent the model from using information of time steps after the prediction time step. We use the same features as for SHAPformer for both the encoder and the

decoder, and no static covariates. Crucially, the model's first layer is a variable selection network, which computes two sets of variable weights – one for the encoder and one for the decoder – that can be interpreted as feature importance values.

**Hyperparameter optimization.** The hyperparameters of the Transformer, TFT and SHAPformer are optimized with bayesian optimization in weights and biases (wandb)[65]. The hyperparameters are given in the Supplementary Methods. The models were trained with Adam[66] and AdamW[67].

## Evaluation metrics
**Forecast evaluation metrics.** The mean absolute error (MAE), root mean squared error (RMSE) and mean absolute percentage error (MAPE) are used to evaluate the forecasts. MAPE is not used on the price dataset, because the price can be zero, where the MAPE is not defined. Letting $y$ be the concatenated target vectors and $\widehat{y}$ the concatenated forecasts, with $M$ entries each, the metrics are defined as follows:

$$\mathrm{MAE}\,(y,\widehat{y}) = \frac{1}{M}\sum_{i=1}^{M}\left|y_i - \widehat{y}_i\right| \quad (7)$$

$$\mathrm{RMSE}\,(y,\widehat{y}) = \sqrt{\frac{1}{M}\sum_{i=1}^{M}\left(y_i - \widehat{y}_i\right)^2} \quad (8)$$

$$\mathrm{MAPE}\,(y,\widehat{y}) = \frac{1}{M}\sum_{i=1}^{M}\left|\frac{y_i - \widehat{y}_i}{y_i}\right| \quad (9)$$

**Explanation evaluation metrics.** The explanations are evaluated by the feature importance error (FIE), local explanation error (LEE) and feature importance instability (FII).
- The FIE is defined as the MAE of the feature importance values shown in Fig. 2A.
- The LEE compares local explanations with local ground truth explanations. A local explanation consists of $N \times H$ SHAP values for $N$ features and forecast length $H$. The LEE is defined as the MAE of the local explanations compared to the ground truth explanations.
- For the FII, we repeat the model training and evaluation five times and calculate the standard deviation of the five feature importance values (visualized as error bars in Figs. 3A, 5A). We then take the mean over all features as the FII.

## Data availability
All three datasets are publicly available, and download links are provided at https://github.com/KIT-IAI/SHAPformer[57]. The synthetic data is published on Zenodo[58]. The load data is originally from open-power-system-data.org[50]. The electricity price data is originally from ENTSO-E[52]. The exogenous weather data is originally from cds.climate.copernicus.eu[51].

## Code availability
SHAPformer is available as a Python package at https://github.com/KIT-IAI/SHAPformer[57].

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

## Acknowledgements

During the preparation of this work, the authors used ChatGPT to improve the clarity and fluency of the written text. After using this tool, the authors reviewed and edited the content as needed and take full responsibility for the content of the published article.

## Author contributions

**M.H.:** Conceptualization, Methodology, Software, Investigation, Writing - original draft, Visualization; **S.P.:** Methodology, Writing - review & editing; **R.M.:** Conceptualization, Supervision, Funding acquisition, Writing - review & editing; **V.H.:** Supervision, Funding acquisition, Writing - review & editing; **B.S.:** Conceptualization, Funding acquisition, Writing - review & editing.

## Funding

**M.H**. discloses support for the research of this work from the Helmholtz Association under the program "Energy System Design", the Helmholtz Association's Initiative and Networking Fund through Helmholtz AI, and the Helmholtz Association's Initiative and Networking Fund on the HAI-CORE@KIT partition. **S.P**. discloses support for the research of this work from the Helmholtz Association's Initiative and Networking Fund through Helmholtz AI. **R.M**. discloses support for the research of this work from the Helmholtz Association under the program "Energy System Design" and the Helmholtz Association's Initiative and Networking Fund through Helmholtz AI. **V.H**. discloses support for the research of this work from the Helmholtz Association under the program "Energy System Design". **B.S**. discloses support for the research of this work from the Helmholtz Association's Initiative and Networking Fund through Helmholtz AI. Open Access funding enabled and organized by Projekt DEAL.

## Competing interests

The authors declare no competing interests.
