## [Transparent Peer Review file · Nature Communications]

Explainable Time-Series Forecasting With Sampling-Free SHAP for Transformers

Corresponding Author: Mr Matthias Hertel

Version 0:

Reviewer comments:

Reviewer #1

(Remarks to the Author)

The manuscript addresses a relevant and timely topic and shows potential for contributing to the field. However, in its current form, it falls short of the scientific and presentation standards required for publication. The work requires major restructuring, deeper methodological justification, and significantly stronger experimental validation. While the core idea is interesting, the current version lacks the clarity, rigor, and completeness necessary for a thorough evaluation. I therefore recommend rejection at this stage, but I encourage the authors to carefully address the comments below and consider resubmitting a substantially revised version in the future.

- **Abstract:** The abstract does not clearly mention the use of Transformer, which are central to the work. Including this information would help clarify the technical focus of the paper.

- **Title:** The title does not indicate that the study focuses on electrical load forecasting. It should be revised to reflect this context more explicitly.

- **Literature Review:** The review lacks sufficient coverage of prior work related to electrical load/power forecasting using Transformers and SHAP methods in the energy domain. Key examples of studies that should be discussed include:

- o “An Informer Model for Very Short-Term Power Load Forecasting”

- o “Interpretable Wind Power Forecasting Combining Seasonal-Trend Representations Learning with Temporal Fusion Transformers Architecture”

- o “Explainable Multi-Step Heating Load Forecasting: Using SHAP Values and Temporal Attention Mechanisms for Enhanced Interpretability”

Including such works would strengthen the contextual foundation and highlight the paper’s novelty more effectively.

- **Research Gap and Contribution:** Based on the above, the manuscript does not clearly define the existing challenges in the field or how the proposed work specifically addresses them. This needs to be articulated more explicitly.

- **Experimental Validation:** For a journal of this caliber, additional experiments on real-world datasets are expected. The current analysis, limited to the German state of Baden-Württemberg, is not sufficient to demonstrate generalizability.

- **Experiment Duration:** The experimental period is quite short (less than three weeks). As a result, some features—such as temperature and month—are less relevant and potentially misleading, as can be observed in Figure 4.

- **Performance Evaluation:** According to Tables 1 and A1, the main advantage of the proposed method appears to be inference time. In terms of RMSE, performance is slightly worse than the baseline Transformer model. Some improvement, or a method to combine the strengths of both approaches, would be expected.

- **Ground truth explanation:** The use of the SHAP Permutation Explainer as a “ground truth” explanation is unclear. This approach measures similarity to SHAP-Permutation attributions, but SHAP itself does not represent a physical ground truth. Without a defined physical reference, this comparison may not be meaningful. Clarification is needed on the rationale and interpretation of these results, mainly from a physical or domain expert perspective.

- **Interpretation of SHAP Values:** The discussion of SHAP results is limited. For instance, in Figure D5, the ‘hour of the day’ feature changes from positive to negative influence. While this is mentioned, there is no accompanying explanation or conclusion. A deeper interpretation of such behavior is necessary to support the claims of interpretability.

To conclude, in its current stage, the paper does not meet the journal’s publication standards and should therefore be rejected. However, if the authors thoroughly address all the comments and substantially improve the manuscript, it may be reconsidered for review in the future.

(Remarks on code availability)

I did not check the code. Please refer to my review above.

Reviewer #2

(Remarks to the Author)

The manuscript introduces SHAPformer, a new approach for computing and interpreting SHAP values that is intended to provide improved robustness and computational efficiency at the program level. The approach is fundamentally relevant, as efficiency and stability are central challenges of modern explainability methods. However, it remains unclear whether the chosen comparison models (Transformer vs. SHAPformer) are methodologically meaningful, given that a Transformer is not a SHAP-based method. The work offers interesting observations, but it lacks sufficient validation and positioning relative to established methods (e.g., GradientSHAP, DeepSHAP, TreeSHAP). Overall, the approach shows potential, but the methodological foundation is not yet adequately developed.

Major comments:

- Justify the choice of the Permutation Explainer as the SHAP baseline by comparing it to alternative SHAP implementations (e.g., DeepSHAP, TreeSHAP) and explaining why this variant was selected over others for time-series data.
- Missing validation: The presented results are difficult to interpret without comparison to established SHAP implementations. Validation of the SHAPformer results against known ground-truth data is missing.
- Provide detailed information on the TransnetBW dataset, including temporal coverage (years), temporal resolution (e.g., hourly), data volume (number of samples), and explicit train/validation/test split dates to enable reproducibility and fair comparison with future work.
- Provide explicit preprocessing details for the TransnetBW dataset: scaling method (e.g., StandardScaler, MinMaxScaler), outlier detection and handling procedures, missing-value imputation strategy, and temporal train/test split dates to ensure reproducibility.
- Quantify SHAPformer's alignment with ground-truth explanations on page 7 using a numerical metric (e.g., error). Consider adding this metric to Figure 2A to facilitate future benchmarking.
- Expand the hyperparameter study description in Appendix B to include: search-space ranges for each hyperparameter in the Bayesian Optimization, total number of optimization trials, training-epoch schedules, early-stopping criteria, and convergence behavior.
- Include a dedicated Limitations section in the Discussion that explicitly names constraints: exponential scaling of attention with feature groups, the model's simplification via monthly aggregation, limited validation on a single real-world dataset, and any assumptions made during synthetic-data generation.
- The advantages of the proposed approach remain too vague. In addition to speed, aspects such as applicability, stability, interpretability, or resource efficiency should be quantified.

Minor comments:

- Introduce the full name Extreme Gradient Boosting (XGBoost) at first mention in the Results section in Table 1 (L84–86).
- Report computational specifications: GPU/CPU type and available memory, and the total number of trainable parameters for each model to enable fair reproduction and performance assessment.
- Figure 4: The red "Load" line is difficult to distinguish due to color choice; the comparison between the SHAP values of the measured values and the target is hard to see. Consider alternative visual representations.
- Clarify the description of TFT's feature importance on line 127: specify "two separate sets (one for past inputs and one for future covariates)" instead of the more generic phrasing "two separate sets of feature importance values."
- Distinguish TFT's encoder and decoder feature importance in Figure 2A by using different colors or line styles for "TFT past" and "TFT future" to improve clarity.
- Standardize terminology across Figures 2–3: consistently use either abbreviated or full model names (e.g., always "Temporal Fusion Transformer" or always "TFT") and use "SHAP" consistently for SHAP values.
- In Equation 2 (L278), format "SHAP" in roman font (SHAP or SHAP) instead of italic to clearly mark it as an acronym.
- Standardize hyphenation throughout the manuscript: consistently choose either "day-of-week", "time-series" (hyphenated) or "day of week", "time series" (unhyphenated).
- Move the inline mathematical expressions on lines 351–354 into a dedicated equation environment to improve clarity and consistency with mathematical typesetting conventions.
- Explicitly label the encoder and decoder regions in Figure 5A to help orient readers who are unfamiliar with Transformer-architecture diagrams.
- Introduce Owen values earlier in the manuscript (e.g., in the Results or Introduction) when first discussing the handling of past load features, rather than defining them only in the Methods section. This is essential for understanding how past load is incorporated into feature-importance computation.
- Distinguish self-attention and cross-attention mechanisms on page 16 (L375–384) with a concise explanation: specify which operates within a single sequence and which fuses information across sequences.

(Remarks on code availability)

Reviewer #3

(Remarks to the Author)

(Remarks on code availability)

Version 1:

Reviewer comments:

Reviewer #2

(Remarks to the Author)

I have no further comments. All of my previous comments have been adequately addressed.

(Remarks on code availability)

Reviewer #3

(Remarks to the Author)

(Remarks on code availability)

Author Response

Matthias Hertel^{1*}, Sebastian Pütz¹, Ralf Mikut¹, Veit Hagenmeyer¹, Benjamin Schäfer¹

¹Institute for Automation and Applied Informatics (IAI), Karlsruhe Institute of Technology (KIT),
Hermann-von-Helmholtz-Platz 1, 76344 Eggenstein-Leopoldshafen, Germany.

*Corresponding author(s). E-mail(s): matthias.hertel@kit.edu;

Contributing authors: sebastian.puetz@kit.edu; ralf.mikut@kit.edu; veit.hagenmeyer@kit.edu;
benjamin.schaefer@kit.edu;

General reply

We sincerely thank the editor and the reviewers for the time and effort invested in the evaluation of our manuscript and for the constructive comments and suggestions. The reviewers' comments have been very valuable in improving the manuscript. In the revised version, we have carefully addressed all comments and suggestions and have made substantial revisions throughout the manuscript. Below, we provide a detailed, point-by-point response to each comment, indicating how the manuscript has been revised accordingly. The comments of the reviewers are written in **blue**, our responses in **black**, and passages from the updated manuscript in **green**. We hope that the revisions satisfactorily address the concerns raised and that the improved manuscript is now suitable for publication.

Reviewer 1

Comment 1.0

The manuscript addresses a relevant and timely topic and shows potential for contributing to the field. However, in its current form, it falls short of the scientific and presentation standards required for publication. The work requires major restructuring, deeper methodological justification, and significantly stronger experimental validation. While the core idea is interesting, the current version lacks the clarity, rigor, and completeness necessary for a thorough evaluation. I therefore recommend rejection at this stage, but I encourage the authors to carefully address the comments below and consider resubmitting a substantially revised version in the future.

Thank you for highlighting the relevance of the topic and the potential of our work, as well as for your constructive comments, which helped us to improve the clarity, rigor and completeness of the work. We respond to your comments one by one.

Comment 1.1

• Abstract: The abstract does not clearly mention the use of Transformer, which are central to the work. Including this information would help clarify the technical focus of the paper.

Thank you for making this observation. We mention the Transformer now in the abstract to clarify the focus of the paper:

We introduce SHAPformer, an accurate, fast and sampling-free explainable time-series forecasting model based on the Transformer architecture.

In addition, we mention early in the Introduction that Transformers are commonly used for time-series forecasting, which motivates our work on using SHAP to explain Transformer-based time-series models:

Across domains, modern forecasting models increasingly rely on complex deep learning architectures [1, 2], including Transformer models for particular datasets [3, 4] and Transformer-based Time Series Foundation Models [5].

Comment 1.2

- **Title:** The title does not indicate that the study focuses on electrical load forecasting. It should be revised to reflect this context more explicitly.

SHAPformer is applicable to any time-series forecasting task and is not designed specifically for electrical load forecasting. We demonstrate the general applicability of SHAPformer in the updated manuscript by evaluating it on electricity price forecasting as an additional use case – see our answer to comment 1.5 below and the updated Results section in the manuscript. Furthermore, additional applications are possible in the future, as discussed in our updated outlook:

SHAPformer can be applied in univariate settings – explaining the prediction based on windows of the past target alone – and in covariate-informed settings, thereby covering a variety of use cases. Given its successful validation on synthetic data and evaluation on two forecasting tasks, we are confident that SHAPformer will generalize well to other use cases.

Comment 1.3

- **Literature Review:** The review lacks sufficient coverage of prior work related to electrical load/power forecasting using Transformers and SHAP methods in the energy domain. Key examples of studies that should be discussed include:
 - o “An Informer Model for Very Short-Term Power Load Forecasting”
 - o “Interpretable Wind Power Forecasting Combining Seasonal-Trend Representations Learning with Temporal Fusion Transformers Architecture”
 - o “Explainable Multi-Step Heating Load Forecasting: Using SHAP Values and Temporal Attention Mechanisms for Enhanced Interpretability”
 Including such works would strengthen the contextual foundation and highlight the paper’s novelty more effectively.

Thank you for suggesting these references, which are indeed relevant for our work. We have added the mentioned references [6], [7] and [8] to the literature review. Note that [6] is the only publication using SHAP in combination with a Transformer-based model, and they do not report runtimes, so we assume that our efficient algorithm to compute SHAP values for Transformers would also be beneficial in their case. The Temporal Fusion Transformer (TFT) used by [7] has limited explanation capabilities, and we compare our approach with TFT in the Results section.

SHAP is the most popular XAI method in the energy sector [9] and has been used to explain electrical load forecasts [10] from models such as multilayer perceptrons [11] and tree-based algorithms [12–15]. SHAP is also used to explain long short-term memories (LSTMs) [8, 16, 17]. Yang et al. [6] use SHAP to explain an Informer [18] model for very short-term power load forecasting and to select features. [...] The Temporal Fusion Transformer (TFT) [19] is a specialized Transformer architecture that computes feature importance values with a dedicated feature selection layer. It is used to analyze the most important features for wind power forecasting [7] and frequency forecasting [20], but unlike SHAP, it does not provide information on the features’ influence on the prediction.

Comment 1.4

- **Research Gap and Contribution:** Based on the above, the manuscript does not clearly define the existing challenges in the field or how the proposed work specifically addresses them. This needs to be articulated more explicitly.

We added two paragraphs describing the research gap and our contribution explicitly:

Despite the growing adoption of Transformer architectures for time-series forecasting, explaining their predictions remains challenging. While SHAP is widely used for model interpretability in the energy domain, existing SHAP algorithms are computationally expensive and rely on sampling or baseline substitutions that can produce unrealistic inputs. As a result, there is currently no efficient and principled method to compute SHAP-based feature contributions for Transformer-based time-series models [10]. This limitation motivates the development of approaches that can provide faithful SHAP explanations for such models without relying on sampling or predefined baselines.

To address this issue, we present a new algorithm for estimating SHAP values for time-series Transformer models based on attention manipulation [21]. Our contributions are as follows:

1. We introduce SHAPformer, a Transformer-based forecasting model that enables the efficient calculation of exact SHAP values by grouping features and applying attention manipulation [21] to exclude feature groups. This approach eliminates the need for sampling or setting baseline values. An overview of the method is shown in Figure 1.
2. We validate SHAPformer’s explanations on a synthetic time-series dataset with known ground-truth explanations, demonstrating its ability to accurately capture the underlying patterns.
3. We showcase the benefits of SHAPformer applied to empirical load data from a transmission system operator (TSO) and day-ahead electricity price data of a market bidding zone. SHAPformer generates explanations in less than one second and provides meaningful local and global insights.

Comment 1.5

- **Experimental Validation:** For a journal of this caliber, additional experiments on real-world datasets are expected. The current analysis, limited to the German state of Baden-Württemberg, is not sufficient to demonstrate generalizability.

We thank the reviewer for this suggestion and added electricity price forecasting as a second use case. In total, we experiment with three datasets: (1) synthetic data used to validate the explanations retrieved with SHAPformer; (2) electrical load data of a transmission system operator (TSO); (3) electricity price data of the DE/LU bidding zone. On all three datasets, SHAPformer matches the forecast accuracy of the compared methods and returns insightful explanations matching the ground truth (which is only available for the synthetic data) and domain knowledge, while achieving a significant speedup compared to other SHAP algorithms. Note that including a large number of further datasets is not feasible, as we require domain knowledge for each dataset to verify the plausibility of explanations. However, with the three datasets investigated, we are confident in SHAPformer’s capabilities and leave additional datasets for future research.

Let us turn to the new electricity price forecasting task investigated. After hyperparameter tuning, we find that SHAPformer achieves the best forecasting performance among all tested models, see Table 1. Meanwhile, the feature importance and feature dependence plots, see Figure 1, align again with our domain understanding. The forecast accuracy and the explanations are described in the updated manuscript as follows:

SHAPformer’s forecast accuracy is comparable to or better than that of the other models reported in Table 1. All evaluated models outperform the persistence baseline, which simply predicts the value from one week earlier. The Transformer achieves the lowest forecast error on the synthetic and load datasets, with SHAPformer close behind with a forecast error that is only about 1 % higher. On the price dataset, SHAPformer is the best model, outperforming TFT and the Transformer by 4.4 % and 7.9 %.

The feature importance and feature dependence plots for the price data, see Figure 1, align again with our domain understanding [22]. Aside from the past price and temporal signals, wind speed is used as an important feature with approximately a negative and linear dependency. Electricity prices are often assumed to linearly increase with residual load, i.e. the load minus must-run-capacity. As wind and solar power, entering our model via the irradiance feature, are typically in the must-run category, they will reduce the price, as we see in our model. The hour-of-day feature resembles the typical price peaks in the morning and the evening, while the day-of-week and holiday features reduce the price primarily on non-working days. The month feature shows no clear pattern; presumably the model reflects high prices associated with specific periods in the training data.

Table 1: Comparison of forecast models. Test root mean squared error (RMSE), training time and inference time on the three datasets.

Model	Forecast error (RMSE)			Training time			Runtime/forecast		
	Synthetic	Load	Price	Synthetic	Load	Price	Synthetic	Load	Price
	-	[MW]	[€/MWh]	[h]	[h]	[h]	[ms]	[ms]	[ms]
Persistence baseline	0.152	652.3	49.51	-	-	-	0.6	1.2	1.2
Linear Regression	0.149	553.7	40.31	0.00	0.00	0.00	0.0	0.3	0.0
XGBoost	0.119	387.0	47.98	0.01	0.00	0.00	0.0	0.1	0.0
TFT	0.059	390.8	34.52	5.20	1.84	0.16	8.4	13.3	5.2
Transformer	0.059	263.1	35.79	0.90	0.26	0.18	11.9	4.7	5.9
SHAPformer	0.060	265.9	33.00	10.55	3.46	0.38	10.8	4.7	3.6

Table 2: Comparison of XAI methods. The feature importance error and local explanation error are evaluated on the synthetic data with ground-truth explanations, and the feature importance instability is defined as the standard deviation of the feature importance values of during experiment repetitions.

XAI method	SHAP values	Runtime/explanation			Feature importance error [%]	Local explanation error	Instability	
		Synthetic [s]	Load [s]	Price [s]			Load	Price
TFT	no	0.01	0.03	0.01	8.04	-	5.57	5.39
KernelSHAP	approx.	252.03	136.65	190.35	11.09	0.063	0.50	0.83
PermutationSHAP	approx.	1124.16	484.34	789.26	7.98	0.056	0.75	0.85
WindowSHAP	approx.	7.84	3.54	5.89	5.41	0.049	0.40	0.73
SHAPformer	exact	21.90	0.60	0.73	1.64	0.016	1.60	1.82

Comment 1.6

- **Experiment Duration:** The experimental period is quite short (less than three weeks). As a result, some features – such as temperature and month – are less relevant and potentially misleading, as can be observed in Figure 4.

We missed to state the experimental period in the manuscript and thank the reviewer for this observation. In fact, the test set for the evaluation of the forecasting metrics reported in Table 1 spans half a year. Figure 4 only shows two exemplary local explanations, while the global explanations presented in Figure 3 are based on evaluations of the entire development and test set (one year in total), in order to cover all seasons. The information on the validation and test splits was updated in the manuscript as follows:

Load dataset: This dataset contains hourly electrical load measurements from the German TSO TransnetBW for the years 2015–2019 [23] and weather data (outside temperature, precipitation) from the Copernicus ERA5 reanalysis model [24]. The first half of 2019 is used for validation and the second half for testing.

We also mention the evaluation period for the global explanations in Figure 3:

With SHAPformer validated on synthetic data, we examine its global explanations for the last 12 months of the load and price datasets, which were not used for model training.

Comment 1.7

- **Performance Evaluation:** According to Tables 1 and A1, the main advantage of the proposed method appears to be inference time. In terms of RMSE, performance is slightly worse than the baseline Transformer model. Some improvement, or a method to combine the strengths of both approaches, would be expected.

Our goal is to combine the strong forecasting performance of the Transformer with a fast algorithm to compute calibrated explanations. As SHAPformer is based on the Transformer architecture, it should in theory have similar forecast accuracy, but our explanation method requires to train the model on masked inputs, which sometimes slightly deteriorates the forecast accuracy. On the synthetic data and the load data, SHAPformer’s RMSE is 1.69 % and 1.06 % higher than that of the Transformer, whereas SHAPformer’s RMSE is better on the electricity price data (see Table 1). We think these small differences are acceptable, given that SHAPformer’s explanations are better calibrated than those of the Transformer explained with KernelSHAP, PermutationSHAP or WindowSHAP (see the improved feature importance error and local explanation error in Table 2), and SHAPformer is up to a factor of 800× faster than existing SHAP methods on the two non-synthetic datasets (see the runtime per explanation reported in Table 2).

We clarified the goal of our work in the Introduction and discuss the pros and cons of SHAPformer in the Discussion:

Thereby, our goal is to combine the strong predictive performance of time-series Transformers with the transparency provided by SHAP.

SHAPformer is designed to combine the strong predictive capabilities of Transformer architectures with an efficient method for computing calibrated SHAP explanations. Because SHAPformer requires training on masked inputs to enable exact attribution, this modification can slightly affect forecasting accuracy compared to a standard Transformer trained solely for prediction. In our experiments, this results in a small increase in RMSE on two datasets, while SHAPformer performs better on the electricity price dataset. We consider these differences acceptable given the substantial gains in interpretability and efficiency: SHAPformer produces explanations that are more closely aligned with ground truth SHAP explanations and reduces the computational cost of explanation generation by up to three orders of magnitude.

A Feature importance and standard deviation across five runs

B SHAPformer dependence plots

Fig. 1: Global explanations on electricity price data. A: Feature importance scores by SHAPformer, a Transformer model explained using different SHAP algorithms, and Temporal Fusion Transformer (TFT). B: Dependence plots from SHAPformer, with feature values on the x-axis and corresponding SHAP values on the y-axis. The strongest interacting variable is indicated by color. For discrete variables, noise was added in the x-direction for visibility reasons.

Thus, SHAPformer represents a trade-off between marginally reduced predictive accuracy and improved explainability and runtime.

Comment 1.8

• Ground truth explanation: The use of the SHAP Permutation Explainer as a “ground truth” explanation is unclear. This approach measures similarity to SHAP-Permutation attributions, but SHAP itself does not represent a physical ground truth. Without a defined physical reference, this comparison may not be meaningful. Clarification is needed on the rationale and interpretation of these results, mainly from a physical or domain expert perspective.

We apologize for the confusion about “ground truth”. Indeed, the SHAP Permutation Explainer often only approximates the model, let alone the data-generating process. Hence, we designed a synthetic dataset for which we know the exact data generating process, i.e. the physical ground truth, including how different features influence the time series. We validate SHAPformer on this dataset, i.e. we re-obtain the known dependencies from the data with only small modifications. This has been clarified in the revised manuscript:

In order to validate the explanations from the different XAI methods presented in Section 2.2, we calculate ground truth SHAP values for the synthetic dataset. This is possible because we have access to the data generation procedure and can therefore explain the true data dependencies with SHAP. As SHAP is model agnostic, it can be used to explain an arbitrary function $f(x)$. Usually, in the context of XAI, f is a machine learning model, but we set f to the data generation process in order to calculate ground-truth explanations. The inputs x to the data generation process consist of the past load, hour of day, day of week, month, holiday, multiplier and two noise features, each represented as a time series of 168 past and – for all inputs except for the load – 168 future values, and the output is the target load of the next week.

We compute ground-truth explanations for the examples in the test set of the synthetic dataset. To do so, we use PermutationSHAP on the data generation process. For a given test example and a permutation, PermutationSHAP computes marginal contributions of the inputs based on subsets of the inputs. The inputs that are absent, i.e. not contained in the subset, are resampled 1000 times following the sampling process described above, and an alternative target load curve is generated using the data generation process and the resampled inputs. These 1000 alternative targets are then averaged, and the averaged values are used as the expected target load curve given the subset of the inputs. From the target load curves generated with different input subsets, marginal contributions and SHAP values are computed with the procedure described in Section 4.1.

While sampling alternative inputs, we make sure that no unrealistic combinations of inputs are generated. In particular, when an input x_1 depends on an input x_2 that is contained in the active subset, we do not resample x_1 . The following dependencies are respected:

- If the day of the week is in the active subset, the hour of day is not resampled (as it can be inferred from the day beginnings and endings).
- If the holiday feature is in the active subset and there is at least one holiday in the example, the hour of day is not resampled (for the same reason).
- If the load of the past week is in the active subset, all calendric information as well as the multiplier of the last week is not resampled (because they affect the load), as well as the load patterns for the workday, Saturday and Sunday (as changing any of these would affect the past load).

Comment 1.9

• **Interpretation of SHAP Values:** The discussion of SHAP results is limited. For instance, in Figure D5, the ‘hour of the day’ feature changes from positive to negative influence. While this is mentioned, there is no accompanying explanation or conclusion. A deeper interpretation of such behavior is necessary to support the claims of interpretability.

We have added a clarification on the interpretation of positive and negative SHAP values:

A positive SHAP value indicates that a feature contributes to increasing the model’s prediction relative to the mean prediction, whereas a negative SHAP value indicates that the feature contributes to decreasing the prediction.

The SHAP values of the hour-of-day feature resemble a typical electricity demand pattern throughout the day. The electricity consumption is low at night, so SHAP values for the hour-of-day feature are negative at night. The electricity demand increases over the day, peaks around noon and shows a second peak in the evening – the same is visible in the SHAP values of the hour-of-day feature. This effect is described in Section 2.3 for local explanations of SHAPformer (Figure 5):

Figure 5 presents two local SHAPformer explanations for forecasts in December. In both cases, the hour of day feature lowers the predictions at night and shows two daily peaks – mirroring the patterns seen in the dependence plots.

Comment 1.10

To conclude, in its current stage, the paper does not meet the journal’s publication standards and should therefore be rejected. However, if the authors thoroughly address all the comments and substantially improve the manuscript, it may be reconsidered for review in the future.

Thank you again for your constructive and detailed feedback. We are convinced that our extensive clarifications, additional interpretations and in particular, the addition of further analysis (price forecasting) addressed your comments as best as possible.

Reviewer 2

Comment 2.0

The manuscript introduces SHAPformer, a new approach for computing and interpreting SHAP values that is intended to provide improved robustness and computational efficiency at the program level. The approach is fundamentally relevant, as efficiency and stability are central challenges of modern explainability methods. However, it remains unclear whether the chosen comparison models (Transformer vs. SHAPformer) are methodologically meaningful, given that a Transformer is not a SHAP-based method. The work offers interesting observations, but it lacks sufficient validation and positioning relative to established methods (e.g., GradientSHAP, DeepSHAP, TreeSHAP). Overall, the approach shows potential, but the methodological foundation is not yet adequately developed.

Thank you for highlighting the relevance of our approach and for your valuable feedback, which helped us to improve the manuscript. You are right that a Transformer is not a SHAP-based method. We have chosen the Transformer as a state-of-the-art forecasting model, and we explain it with three SHAP algorithms: the model-agnostic KernelSHAP and PermutationSHAP, and WindowSHAP that is specific for time series. In addition, we compare SHAPformer's predictive performance to that of a Linear Regression, XGBoost and Temporal Fusion Transformer (Table 1), as well as the feature importance with TFT (Fig. 2A). We address your other comments one by one, including the comment about positioning our method relative to other SHAP algorithms. The reasoning for comparing to the Transformer was included as follows:

As SHAPformer is based on the Transformer architecture, we first compare its predictive performance to a Transformer forecasting model and several baselines. Explanations for the Transformer are then generated with three SHAP algorithms: (1) KernelSHAP, which estimates SHAP values by fitting a linear model on predictions based on perturbed inputs; (2) PermutationSHAP, which perturbs inputs using background data under the assumption of feature independence and estimates SHAP values directly from the model predictions on the perturbed samples; and (3) a variant of WindowSHAP, which splits the past target into seven windows (one for each of the past seven days) and uses one extra window for each exogenous variable. SHAP values are then computed per window, instead of per feature, by jointly sampling all features within a window.

Major comments:

Comment 2.1

- Justify the choice of the Permutation Explainer as the SHAP baseline by comparing it to alternative SHAP implementations (e.g., DeepSHAP, TreeSHAP) and explaining why this variant was selected over others for time-series data.

We thank the reviewer for this helpful suggestion and have expanded the manuscript to discuss existing SHAP implementations in more detail in Section 1. In particular, we now provide an overview of commonly used SHAP algorithms, including KernelSHAP, PermutationSHAP, TreeSHAP, DeepSHAP, GradientSHAP, and time-series-specific methods such as TimeSHAP, WindowSHAP, and ShapTime.

Several of these approaches are not directly applicable to our model. TreeSHAP is designed specifically for tree-based models and therefore cannot be applied to Transformer architectures. DeepSHAP is implemented only for a limited set of neural network operations and currently does not support the LayerNorm operation, which is an essential component of Transformer models. We also attempted to apply GradientSHAP, but our model contains categorical inputs that are first mapped to embeddings. Because the embedding operation is not differentiable with respect to the categorical indices, GradientSHAP cannot be applied directly at the input level. In principle, explanations could be computed on the level of embeddings instead of the original categorical inputs, but this would require a substantial restructuring of the model and explanation pipeline. Moreover, GradientSHAP relies on sampling from background data, similar to other SHAP variants, and therefore shares the associated computational costs and potential off-manifold issues.

In contrast, KernelSHAP and PermutationSHAP are model-agnostic methods that can be applied to arbitrary models. Following the reviewer's suggestion, we have now added KernelSHAP to our experimental comparison in addition to PermutationSHAP. Furthermore, we clarified that the method previously referred to as a Custom Masker corresponds to a variant of WindowSHAP adapted for exogenous variables; this terminology has been updated in the manuscript. ShapTime and TimeSHAP follow very similar approaches as WindowSHAP and are therefore not evaluated additionally.

Overall, these revisions clarify the relationship between SHAPformer and existing SHAP implementations and provide a more comprehensive justification of the baselines used in our experiments.

A new paragraph was introduced in Section 1, which gives an overview of existing SHAP algorithms:

The following list covers popular SHAP algorithms, including three approaches tailored to time-series models [25–27]:

- KernelSHAP [28] estimates SHAP values for a given instance by fitting a weighted local linear regression to outputs of the model evaluated on perturbed samples, where absent features are sampled from background data.
- PermutationSHAP [28] estimates SHAP values by repeatedly permuting features and computing marginal feature contributions based on sampling absent features from background data.
- TreeSHAP [29] is specific to tree-based models and leverages the tree structure to estimate SHAP values efficiently.
- DeepSHAP [28] approximates SHAP values for neural networks by computing the difference between the model output for a given input and its output for multiple background samples, and propagating these differences backward through the network using DeepLIFT [30] contribution rules. The final attributions are obtained by averaging over the background samples.
- GradientSHAP is a SHAP algorithm for differentiable models that is based on Expected Gradients [31]. It iterates over background data and evaluates the gradient of the model output with respect to the input at random locations between the model input and randomly chosen background samples.
- TimeSHAP [25] is specific to time-series data and estimates SHAP values for selected time points and features by using KernelSHAP with perturbed inputs where absent values are replaced with the respective feature’s mean value.
- WindowSHAP [26] and ShapTime [27] are two SHAP algorithms for time series, which estimate SHAP values for time windows instead of individual time points to reduce the number of model evaluations.

Comment 2.2

- **Missing validation:** The presented results are difficult to interpret without comparison to established SHAP implementations. Validation of the SHAPformer results against known ground-truth data is missing.

We compare SHAPformer with three established SHAP implementations – KernelSHAP, PermutationSHAP and WindowSHAP – in the new Table 2. SHAPformer calculates explanations much faster on the load and price datasets than the alternative SHAP algorithms. On the synthetic data, which involves more covariates, SHAPformer suffers from its runtime growing exponentially with the number of covariates – this can be improved substantially in the future by calculating approximate SHAP values instead of exact SHAP values. Next to runtime, we evaluate the feature importance error on the synthetic data, which is the only dataset where ground-truth explanations are available. We find that SHAPformer resembles the ground truth better than the existing SHAP algorithms. The same holds for the local explanation error, which compares the retrieved local explanations with the ground truth.

Comment 2.3

- **Provide detailed information on the TransnetBW dataset, including temporal coverage (years), temporal resolution (e.g., hourly), data volume (number of samples), and explicit train/validation/test split dates to enable reproducibility and fair comparison with future work.**

The requested information was added to the manuscript for all three datasets as follows:

We evaluate SHAPformer on three datasets:

- **Synthetic dataset:** This dataset exhibits daily, weekly, and annual seasonality with dependencies on exogenous covariates (holidays and a multiplier), for which ground-truth explanations are available. The details of the data generation are presented in Section 4.3. 100,000 samples are used for training and 10,000 each for validation and testing.
- **Load dataset:** This dataset contains hourly electrical load measurements from the German TSO TransnetBW for the years 2015–2019 [23] and weather data (outside temperature, precipitation) from the Copernicus ERA5 reanalysis model [24]. The first half of 2019 is used for validation and the second half for testing. The load data is standardized to zero mean and unit variance.
- **Price dataset:** This dataset contains day-ahead electricity prices from the DE/LU bidding zone, collected from the ENTSO-E Transparency platform [32] in hourly resolution for October 2020 until December 2025, and weather data (wind speed, solar irradiance) from the Copernicus ERA5 reanalysis model [24]. The year 2024 is used for validation and the year 2025 for testing. The price data is standardized to zero mean and unit variance.

Comment 2.4

- **Provide explicit preprocessing details for the TransnetBW dataset: scaling method (e.g., StandardScaler, MinMaxScaler), outlier detection and handling procedures, missing-value imputation strategy, and temporal train/test split dates to ensure reproducibility.**

Information about the scaling method and the temporal splits was added (see Comment 2.3 above). The datasets contain no outliers or missing values, so no handling of such cases is needed.

Comment 2.5

- Quantify SHAPformer’s alignment with ground-truth explanations on page 7 using a numerical metric (e.g., error). Consider adding this metric to Figure 2A to facilitate future benchmarking.

This is indeed a good idea, and we thank the reviewer for this suggestion. We introduce two explanation error metrics – one on the level of global feature importance, and the other on the level of local explanations. The metrics are added to the comparison of explanation methods in Table 2, and introduced and discussed in the manuscript as follows:

We calculate the feature importance error, defined as the mean absolute error of the feature importance values shown in Figure 2A with respect to the ground truth, and report it in Table 2. We find that SHAPformer resembles the ground truth feature importance well, deviating only by 1.64 percentage points, whereas the feature importance error of the second-best WindowSHAP is more than three times larger. Since ground-truth explanations are available for all samples in the synthetic dataset, they can be used to evaluate the local explanations of a SHAP method as well. Table 2 reports the local explanation error, defined as the mean absolute error of the local explanations averaged across all samples, forecast horizons and feature groups. Again, we find that SHAPformer resembles the ground truth best, resulting in a three to four times lower local explanation error than the other SHAP algorithms.

Comment 2.6

- Expand the hyperparameter study description in Appendix B to include: search-space ranges for each hyperparameter in the Bayesian Optimization, total number of optimization trials, training-epoch schedules, early-stopping criteria, and convergence behavior.

The hyperparameter search space was added to Table B3, and the description of the hyperparameter optimization extended as follows:

The hyperparameters of the Temporal Fusion Transformer, the Transformer and SHAPformer on synthetic data were optimized using Bayesian Optimization and Weights and Biases (wandb) [33]. The search-space ranges and the selected hyperparameters are reported in Table 3. In total, 100 training runs were performed for each model type. Each model was trained for up to 100 epochs using early stopping with a patience of 10 epochs based on the validation loss. For SHAPformer on load data, the same hyperparameters were used as for the Transformer, but with a lower learning rate, which stabilizes the masked training. Adam [34] and AdamW [35] were used as training algorithms, using the mean squared error loss function.

Comment 2.7

- Include a dedicated Limitations section in the Discussion that explicitly names constraints: exponential scaling of attention with feature groups, the model’s simplification via monthly aggregation, limited validation on a single real-world dataset, and any assumptions made during synthetic-data generation.

We added a paragraph on limitations to the Discussion:

Despite its advantages, SHAPformer has several limitations. First, feature importance and local explanations can vary across training runs, as reflected by the instability metric in Table 2. This metric measures the standard deviation of the feature importance values over five model training and evaluation runs, and the results show that SHAPformer’s feature importance deviates by one to two percentage points. Second, the computational complexity of SHAPformer scales exponentially with the number of feature groups, which limits its applicability when many groups are considered. Future work could address this issue by estimating approximate SHAP values based on a subset of feature groups or permutations. Third, the empirical evaluation is limited to one synthetic dataset with known ground-truth explanations and two real-world datasets since each dataset requires domain understanding to verify explanations, and we leave the evaluation on more datasets to future work. Finally, the synthetic benchmark relies on several assumptions: the ground-truth explanations are defined using SHAP values derived from the data-generating process, they assume that the learned model faithfully represents this process, and they respect dependencies between features (e.g., the hour-of-day can be inferred from the day-of-week feature which indicates day changeovers). While these assumptions allow controlled evaluation, faithful explanations for models that do not follow these assumptions will differ from the ground truth.

Comment 2.8

- The advantages of the proposed approach remain too vague. In addition to speed, aspects such as applicability, stability, interpretability, or resource efficiency should be quantified.

We thank the reviewer for this suggestion and have revised the manuscript to clarify and better quantify the advantages of SHAPformer beyond runtime:

- **Applicability:** We now demonstrate SHAPformer’s applicability on three datasets: a synthetic dataset with known ground-truth explanations and two real-world datasets from different forecasting tasks (electrical load and electricity price). As SHAPformer is based on a Transformer forecasting architecture, the method is generally applicable to a wide range of time-series forecasting problems, including both univariate and covariate-informed settings.
- **Stability:** SHAPformer is a sampling-free method and therefore produces deterministic explanations for a given trained model. However, explanations can still vary across different training runs due to stochasticity in model training. To quantify this effect, we introduce a new stability metric that evaluates the variability of feature importance and local explanations across multiple runs. We find that SHAP-based explanations are more stable than TFT’s explanations, but SHAPformer is less stable than the Transformer explained by PermutationSHAP or WindowSHAP – this is likely due to the fact that the Transformer consistently focusses on the hour-of-day feature whereas SHAPformer distributes the feature importance more equally.
- **Interpretability:** We introduce two quantitative metrics—feature importance error and local explanation error—on the synthetic dataset with ground-truth explanations. These metrics allow us to directly assess how well the explanations recover the known data-generating relationships. In both cases, SHAPformer shows substantially lower error than existing SHAP algorithms (see Table 2).
- **Resource efficiency:** SHAPformer has great runtime advantages over existing SHAP algorithms, thereby allowing to use less compute resources. We have not noted any memory issues with any of the explanation methods, so we do not evaluate memory usage.

Together with the reported runtime improvements, these additions provide a clearer and more comprehensive evaluation of the advantages of the proposed approach. These aspects are included in the updated Discussion:

SHAPformer is designed to combine the strong predictive capabilities of Transformer architectures with an efficient method for computing calibrated SHAP explanations. Because SHAPformer requires training on masked inputs to enable exact attribution, this modification can slightly affect forecasting accuracy compared to a standard Transformer trained solely for prediction. In our experiments, this results in a small increase in RMSE on two datasets, while SHAPformer performs better on the electricity price dataset. We consider these differences acceptable given the substantial gains in interpretability and efficiency: SHAPformer produces explanations that are more closely aligned with ground truth SHAP explanations and reduces the computational cost of explanation generation by up to three orders of magnitude. Thus, SHAPformer represents a trade-off between marginally reduced predictive accuracy and improved explainability and runtime.

SHAPformer can be applied in univariate settings – explaining the prediction based on windows of the past target alone – and in covariate-informed settings, thereby covering a variety of use cases.

Minor comments:

Comment 2.9

- Introduce the full name Extreme Gradient Boosting (XGBoost) at first mention in the Results section in Table 1 (L84–86).

To keep Table 1 within the text width, we keep the established short name “XGBoost”, but we introduce the full name at the first mention in the text:

On the non-synthetic datasets, SHAPformer outperforms Linear Regression, Extreme Gradient Boosting (XGBoost) [36], and TFT.

Comment 2.10

- Report computational specifications: GPU/CPU type and available memory, and the total number of trainable parameters for each model to enable fair reproduction and performance assessment.

The information about the hardware was added as follows. The number of trainable parameters is defined by the hyperparameters given in Table B3 for reproducibility.

All runtimes are evaluated on a machine with an AMD EPYC 7402P processor, 128 GB RAM and a 3090 RTX GPU with 24 GB VRAM.

Comment 2.11

- Figure 4: The red “Load” line is difficult to distinguish due to color choice; the comparison between the SHAP values of the measured values and the target is hard to see. Consider alternative visual representations.

We reduced the alpha value of the stacked SHAP values, so that the prediction (which we changed from a red line to a black line) becomes better visible:

Fig. 2: Local explanation from SHAPformer for the electrical load on December 17, 2019, 22:00.

Comment 2.12

- Clarify the description of TFT’s feature importance on line 127: specify “two separate sets (one for past inputs and one for future covariates)” instead of the more generic phrasing “two separate sets of feature importance values.”

We clarified what are the two separate sets:

TFT is more difficult to interpret, as it produces separate sets of feature importance values for past and future features.

Comment 2.13

- Distinguish TFT’s encoder and decoder feature importance in Figure 2A by using different colors or line styles for “TFT past” and “TFT future” to improve clarity.

We changed the colors for “TFT past” and “TFT future” to dark blue and light blue, to make them distinguishable, yet indicate with the blue color that they belong to the same model.

Comment 2.14

- Standardize terminology across Figures 2–3: consistently use either abbreviated or full model names (e.g., always “Temporal Fusion Transformer” or always “TFT”) and use “SHAP” consistently for SHAP values.

We standardized the naming of the models in the Figures. Regarding SHAP and SHAP values, remember that “SHAP” is the explanation method and “SHAP values” are the outputs of the method (i.e, the resulting explanations). We therefore label the local and global explanations with “SHAP values” to distinguish them from the method “SHAP”.

Comment 2.15

- In Equation 2 (L278), format “SHAP” in roman font (SHAP or SHAP) instead of italic to clearly mark it as an acronym.

“SHAP” is now introduced as an operator name:

$$\text{SHAP}(v_i) = \sum_{S \subseteq V \setminus \{v_i\}} \frac{(n-1-|S|)! \cdot |S|!}{n!} \cdot (f(S \cup \{v_i\}) - f(S)) \quad (1)$$

Comment 2.16

- Standardize hyphenation throughout the manuscript: consistently choose either “day-of-week”, “time-series” (hyphenated) or “day of week”, “time series” (unhyphenated).

To the best of our knowledge, the correct spelling is unhyphenated for compound nouns (a time series, a day of (the) week) and hyphenated for compound adjectives describing a noun (time-series forecasting, the day-of-week feature). We checked the entire manuscript for correctness.

Comment 2.17

- Move the inline mathematical expressions on lines 351–354 into a dedicated equation environment to improve clarity and consistency with mathematical typesetting conventions.

We moved the mathematical expressions as suggested:

More formally, there is a query vector q and key vectors k_1 to k_n . With each key vector k_i there is a value vector u_i associated. The attention score a_i of a key vector k_i is computed as

$$a_i = \frac{q^T k_i}{\sqrt{d}},$$

where d is the vector dimension. The attention scores a_i to a_n are then soft-maxed in order to compute attention weights α_1 to α_n which sum up to one:

$$\alpha_i = \frac{\exp(a_i)}{\sum_{j=1}^n \exp(a_j)}$$

The output o of the attention mechanism is finally computed as

$$o = \sum_{i=1}^n \alpha_i \cdot u_i.$$

Comment 2.18

- Explicitly label the encoder and decoder regions in Figure 5A to help orient readers who are unfamiliar with Transformer-architecture diagrams.

We updated the figure accordingly, see Figure 3.

Comment 2.19

- Introduce Owen values earlier in the manuscript (e.g., in the Results or Introduction) when first discussing the handling of past load features, rather than defining them only in the Methods section. This is essential for understanding how past load is incorporated into feature-importance computation.

We now mention the computation of Owen values in the Introduction:

[...] we present a new algorithm for estimating SHAP values for time-series Transformer models based on attention manipulation [21] and the Owen formula [37] to respect feature correlations.

However, we leave the formula in the Methodology section, so that it gets introduced after the SHAP formula it builds on (Formula (2)).

Comment 2.20

- Distinguish self-attention and cross-attention mechanisms on page 16 (L375–384) with a concise explanation: specify which operates within a single sequence and which fuses information across sequences.

We added an explanation of the self-attention and cross-attention mechanisms:

The self-attention in the encoder operates within the past time steps. [...] The self-attention in the decoder operates within the future time steps, whereas the cross-attention fuses information from the encoder output into the decoder representation of future time steps.

Fig. 3: SHAPformer architecture with labeled encoder and decoder regions.

Reviewer 3

Comment 3.0

We thank you for contributing to the review of our manuscript as part of the co-reviewing initiative and appreciate the time and effort invested in providing thoughtful feedback. Your comments have been valuable in helping us improve the clarity and quality of the manuscript.

References

- [1] Lim B, Zohren S. Time-series forecasting with deep learning: a survey. *Philosophical Transactions of the Royal Society A: Mathematical, Physical and Engineering Sciences*. 2021 Feb;379(2194):20200209. <https://doi.org/10.1098/rsta.2020.0209>.
- [2] Su L, Zuo X, Li R, Wang X, Zhao H, Huang B. A systematic review for transformer-based long-term series forecasting. *Artificial Intelligence Review*. 2025 Jan;58(3):80. <https://doi.org/10.1007/s10462-024-11044-2>.
- [3] Hertel M, Beichter M, Heidrich B, Neumann O, Schäfer B, Mikut R, et al. Transformer training strategies for forecasting multiple load time series. *Energy Informatics*. 2023 Oct;6(1):20. <https://doi.org/10.1186/s42162-023-00278-z>.
- [4] Giacomazzi E, Haag F, Hopf K. Short-Term Electricity Load Forecasting Using the Temporal Fusion Transformer: Effect of Grid Hierarchies and Data Sources. In: *Proceedings of the 14th ACM International Conference on Future Energy Systems*. Orlando FL USA: ACM; 2023. p. 353–360. Available from: <https://dl.acm.org/doi/10.1145/3575813.3597345>.
- [5] Kottapalli SRK, Hubli K, Chandrashekhara S, Jain G, Hubli S, Botla G, et al.: Foundation Models for Time Series: A Survey. Available from: <https://arxiv.org/abs/2504.04011v1>.
- [6] Yang Z, Li J, Wang H, Liu C. An Informer Model for Very Short-Term Power Load Forecasting. *Energies*. 2025 Jan;18(5):1150. <https://doi.org/10.3390/en18051150>.
- [7] Niu Z, Han X, Zhang D, Wu Y, Lan S. Interpretable wind power forecasting combining seasonal-trend representations learning with temporal fusion transformers architecture. *Energy*. 2024 Oct;306:132482. <https://doi.org/10.1016/j.energy.2024.132482>.
- [8] Neubauer A, Brandt S, Kriegel M. Explainable multi-step heating load forecasting: Using SHAP values and temporal attention mechanisms for enhanced interpretability. *Energy and AI*. 2025 May;20:100480. <https://doi.org/10.1016/j.egyai.2025.100480>.
- [9] Machlev R, Heistrene L, Perl M, Levy KY, Belikov J, Mannor S, et al. Explainable Artificial Intelligence (XAI) techniques for energy and power systems: Review, challenges and opportunities. *Energy and AI*. 2022 Aug;9:100169. <https://doi.org/10.1016/j.egyai.2022.100169>.
- [10] Baur L, Ditschuneit K, Schambach M, Kaymakci C, Wollmann T, Sauer A. Explainability and Interpretability in Electric Load Forecasting Using Machine Learning Techniques – A Review. *Energy and AI*. 2024 May;16:100358. <https://doi.org/10.1016/j.egyai.2024.100358>.
- [11] Bolstad DA, Cali U, Kuzlu M, Halden U. Day-ahead Load Forecasting using Explainable Artificial Intelligence. In: *2022 IEEE Power & Energy Society Innovative Smart Grid Technologies Conference (ISGT)*; 2022. p. 1–5. Available from: <https://ieeexplore.ieee.org/document/9817538>.
- [12] Li M, Wang Y. Power load forecasting and interpretable models based on GS_XGBoost and SHAP. *Journal of Physics: Conference Series*. 2022 Feb;2195(1):012028. <https://doi.org/10.1088/1742-6596/2195/1/012028>.
- [13] Lee YG, Oh JY, Kim D, Kim G. SHAP Value-Based Feature Importance Analysis for Short-Term Load Forecasting. *Journal of Electrical Engineering & Technology*. 2023 Jan;18(1):579–588. <https://doi.org/10.1007/s42835-022-01161-9>.
- [14] Han Y, Sha X, Grover-Silva E, Michiardi P. On the impact of socio-economic factors on power load forecasting. In: *2014 IEEE International Conference on Big Data (Big Data)*; 2014. p. 742–747. Available from: <https://ieeexplore.ieee.org/document/7004299>.
- [15] Wang J, Yu B, Chen X, Dai G, Dai G, Liu W, et al. An interpretable short-term electrical load forecasting model based on SHapley Additive exPlanations—A case study in Haidian, Beijing. *Electric Power Systems Research*. 2025 Oct;247:111769. <https://doi.org/10.1016/j.epsr.2025.111769>.

- [16] Wu K, Gu J, Meng L, Wen H, Ma J. An explainable framework for load forecasting of a regional integrated energy system based on coupled features and multi-task learning. *Protection and Control of Modern Power Systems*. 2022 Jun;7(1):24. <https://doi.org/10.1186/s41601-022-00245-y>.
- [17] Henriksen E, Halden U, Kuzlu M, Cali U. Electrical Load Forecasting Utilizing an Explainable Artificial Intelligence (XAI) Tool on Norwegian Residential Buildings. In: *2022 International Conference on Smart Energy Systems and Technologies (SEST)*; 2022. p. 1–6. Available from: <https://ieeexplore.ieee.org/document/9898500>.
- [18] Zhou H, Zhang S, Peng J, Zhang S, Li J, Xiong H, et al. Informer: Beyond Efficient Transformer for Long Sequence Time-Series Forecasting. *Proceedings of the AAAI Conference on Artificial Intelligence*. 2021 May;35(12):11106–11115. Number: 12. <https://doi.org/10.1609/aaai.v35i12.17325>.
- [19] Lim B, Arik SO, Loeff N, Pfister T. Temporal Fusion Transformers for interpretable multi-horizon time series forecasting. *International Journal of Forecasting*. 2021 Oct;37(4):1748–1764. <https://doi.org/10.1016/j.ijforecast.2021.03.012>.
- [20] Pütz S, El Ashhab H, Hertel M, Mikut R, Götz M, Hagenmeyer V, et al. Feasibility of Forecasting Highly Resolved Power Grid Frequency Utilizing Temporal Fusion Transformers. In: *Proceedings of the 15th ACM International Conference on Future and Sustainable Energy Systems. e-Energy '24*. New York, NY, USA: Association for Computing Machinery; 2024. p. 447–453. Available from: <https://dl.acm.org/doi/10.1145/3632775.3661963>.
- [21] Deiseroth B, Deb M, Weinbach S, Brack M, Schramowski P, Kersting K. ATMAN: Understanding Transformer Predictions Through Memory Efficient Attention Manipulation. *Advances in Neural Information Processing Systems*. 2023 Dec;36:63437–63460.
- [22] Trebbien J, Rydin Gorjão L, Praktiknjo A, Schäfer B, Witthaut D. Understanding electricity prices beyond the merit order principle using explainable AI. *Energy and AI*. 2023 Jul;13:100250. <https://doi.org/10.1016/j.egyai.2023.100250>.
- [23] Wiese F, Schlecht I, Bunke WD, Gerbaulet C, Hirth L, Jahn M, et al. Open Power System Data – Frictionless data for electricity system modelling. *Applied Energy*. 2019 Feb;236:401–409. <https://doi.org/10.1016/j.apenergy.2018.11.097>.
- [24] Copernicus Climate Change Service.: Climate and energy indicators for Europe from 1979 to present derived from reanalysis. ECMWF. Available from: <https://cds.climate.copernicus.eu/doi/10.24381/cds.4bd77450>.
- [25] Bento J, Saleiro P, Cruz AF, Figueiredo MAT, Bizarro P. TimeSHAP: Explaining Recurrent Models through Sequence Perturbations. In: *Proceedings of the 27th ACM SIGKDD Conference on Knowledge Discovery & Data Mining. KDD '21*. New York, NY, USA: Association for Computing Machinery; 2021. p. 2565–2573. Available from: <https://doi.org/10.1145/3447548.3467166>.
- [26] Nayebi A, Tipirneni S, Reddy CK, Foreman B, Subbian V. WindowSHAP: An efficient framework for explaining time-series classifiers based on Shapley values. *Journal of Biomedical Informatics*. 2023 Aug;144:104438. <https://doi.org/10.1016/j.jbi.2023.104438>.
- [27] Zhang Y, Sun Q, Qi D, Liu J, Ma R, Petrosian O. ShapTime: A General XAI Approach for Explainable Time Series Forecasting. In: Arai K, editor. *Intelligent Systems and Applications*. Cham: Springer Nature Switzerland; 2024. p. 659–673.
- [28] Lundberg SM, Lee SI. A Unified Approach to Interpreting Model Predictions. In: *Advances in Neural Information Processing Systems*. vol. 30. Canada; 2017. p. 4768 – 4777. Available from: <https://dl.acm.org/doi/proceedings/10.5555/3295222>.
- [29] Lundberg SM, Erion G, Chen H, DeGrave A, Prutkin JM, Nair B, et al. From local explanations to global understanding with explainable AI for trees. *Nature Machine Intelligence*. 2020 Jan;2(1):56–67. <https://doi.org/10.1038/s42256-019-0138-9>.
- [30] Shrikumar A, Greenside P, Kundaje A. Learning important features through propagating activation differences. In: *Proceedings of the 34th International Conference on Machine Learning - Volume 70. ICML'17*. Sydney, NSW,

- Australia: JMLR.org; 2017. p. 3145–3153. Available from: <https://dl.acm.org/doi/10.5555/3305890.3306006>.
- [31] Erion G, Janizek JD, Sturmfels P, Lundberg SM, Lee SI. Improving performance of deep learning models with axiomatic attribution priors and expected gradients. *Nature Machine Intelligence*. 2021 Jul;3(7):620–631. <https://doi.org/10.1038/s42256-021-00343-w>.
- [32] ENTSO-E.: Transparency platform. Available from: <https://transparency.entsoe.eu>.
- [33] Biewald L.: Experiment Tracking with Weights and Biases. Available from: wandb.com.
- [34] Kingma DP, Ba J.: Adam: A Method for Stochastic Optimization. *arXiv*. ArXiv:1412.6980 [cs]. Available from: <http://arxiv.org/abs/1412.6980>.
- [35] Loshchilov I, Hutter F.: Decoupled Weight Decay Regularization. *arXiv*. ArXiv:1711.05101 [cs, math]. Available from: <http://arxiv.org/abs/1711.05101>.
- [36] Chen T, Guestrin C. XGBoost: A Scalable Tree Boosting System. In: *Proceedings of the 22nd ACM SIGKDD International Conference on Knowledge Discovery and Data Mining*; 2016. p. 785–794. ArXiv:1603.02754 [cs]. Available from: <http://arxiv.org/abs/1603.02754>.
- [37] Owen G. Values of Games with a Priori Unions. In: Henn R, Moeschlin O, editors. *Mathematical Economics and Game Theory*. Berlin, Heidelberg: Springer; 1977. p. 76–88.